



# Modal dynamics of structures with bladed isotropic rotors and its complexity for 2-bladed rotors

Morten Hartvig Hansen

Dept. of Wind Energy, Technical University of Denmark, Frederiksborgvej 399, 4000 Roskilde, Denmark

*Correspondence to:* Morten H. Hansen (mhha@dtu.dk)

**Abstract.** The modal dynamics of structures with bladed isotropic rotors is analyzed using Hill's method. First, analytical derivation of the periodic system matrix shows that isotropic rotors with more than two blades can be represented by an exact Fourier series with 3/rev as the highest order. For 2-bladed rotors, the inverse mass matrix has an infinite Fourier series with harmonic components of decreasing norm, thus the system matrix can be approximated by a truncated Fourier series of

predictable accuracy. Second, a novel method for automatically identifying the principal solutions of Hill's eigenvalue problem is introduced. The corresponding periodic eigenvectors can be used to compute symmetric and anti-symmetric components of the 2-bladed rotor motion, and the additional forward and backward whirling components for rotors with more than two blades. Finally, the generic methods are used on a simple wind turbine model consisting of three degrees of freedom for each blade and seven degrees of freedom for the nacelle and drivetrain. The modal dynamics of a 3-bladed 10MW turbine from

previous studies is recaptured. Removing one blade, the larger and higher harmonic terms in the system matrix lead to resonant modal couplings for the 2-bladed turbine that do not exist for the 3-bladed turbine, and that excitation of a single mode of a 2-bladed turbine leads to responses at several frequencies in both the ground-fixed and rotating blade frames of reference which complicates the interpretation of simulated or measured turbine responses.

## 1 Introduction

A fundamental understanding of the modal dynamics of structures with bladed rotors is relevant for the design and analysis of wind turbines, helicopters, and other rotating machinery because their vibrational responses are composed of their structural modes. It is important to understand how these modes are excited by resonances or aeroelastic instabilities, i.e., at which frequencies and where on the structure or rotor the individual modes can be excited. Such knowledge is not only necessary for the interpretation of design simulations but also for the understanding of real measurements.

The modal dynamics of 3-bladed turbines is well understood, also including the interaction with aerodynamic forces and a controller. For isotropic rotors, the Coleman transformation (Coleman & Feingold, 1958) is often used to transform the periodic system into a time-invariant system and then solving the associated eigenvalue problem with the blade motion described by the multi-blade coordinates (Hansen, 2003; Van Engelen & Braam, 2004; Hansen, 2004, 2007; Riziotis et al., 2008; Bir, 2008; Skjoldan & Hansen, 2009; Bergami & Hansen, 2016). The turbine modes may either be dominated by vibrations of the rotor

support structure and drivetrain, e.g. tower bending and shaft torsion modes, or by blade vibrations which herein are called the





*rotor modes*. The rotor modes of the 3-bladed isotropic rotor consists of a symmetric mode and two whirling modes, where the order of blade vibration describes a backward (regressive) and a forward (progressive) whirling direction relative to the rotor rotation. Due to the anisotropy of the rotor support, the rotor modes are not "pure" meaning that for example a backward whirling mode will also contain symmetric and forward whirling components when observed from the rotating blade frame of

reference. The modal frequency obtained from the eigenvalue problem describes the frequency observed from the ground-fixed frame of reference in which the multi-blade coordinates describes the rotor motion. In the rotating blade frame, the symmetric rotor response will be observable at the same frequency but the backward and forward whirling components of a rotor mode will be shifted by +1/rev and -1/rev, respectively. Since a rotor mode is not pure, its response may therefore be observable at three different frequencies in a signal measured on the blade (Hansen, 2003, 2007).

For anisotropic 3-bladed rotors, Floquet theory or Hill's method is needed to obtain an eigenvalue problem which leads to the periodic eigenvectors of the principal eigenvalue solutions (Skjoldan & Hansen, 2009; Bottasso & Cacciola, 2015; Skjoldan, 2009; Skjoldan & Hansen, 2009). To handle the frequency indeterminacy of the periodic eigenvalue solutions from these methods, Skjoldan & Hansen (2009) suggest to select the principal solution such that the harmonic components on the ground-fixed degrees of freedom are minimized. Bottasso & Cacciola (2015) introduce the concept of *modal participation factors*

in which the norm of the individual harmonic components of a periodic eigenvector determines how much the particular component contributes to the response of the particular mode. They also introduce the concept of *periodic Campbell diagrams* to plot the frequencies of these harmonic components along with the principal frequency. All studies show that periodic mode shapes of turbines with 3-bladed anisotropic rotors can contain harmonic components which frequencies can be shifted more than ±1/rev from the principal modal frequency. The size of these higher harmonic components depend on the size of rotor

anisotropy.

The modal analysis of structures with 2-bladed rotors is complicated by the strong periodicity of the system. The use of Floquet theory or Hill's method is unavoidable, unless both rotor and support structure are isotropic with respect to rotation (Coleman & Feingold, 1958). Early studies (Warmbrodt & Friedmann, 1980; Wendell, 1982; Kirchgäßner, 1984) have used Floquet theory to investigate the aeroelastic stability of 2-bladed rotors without focusing on their modal dynamics. How-

ever, Kirchgäßner (1984) introduces the concept of *dominant eigenfrequencies* for the harmonic components of the Floquet solutions with the largest magnitude in the corresponding eigenvector, and he plotted the frequencies of all harmonic components with magnitudes larger than a threshold relative to the dominant component in a periodic Campbell diagram similar to Bottasso & Cacciola (2015). A later study by Stol et al. (2002) considers the dynamic stability of a teetered 2-bladed turbine using Floquet theory on a model with up to seven degrees of freedom. Their focus is mainly on the parametric excitation of the

system and less on its modal dynamics. Recent studies of 2-bladed turbines have focused on their aero-servo-elastic control (Solingen, 2015; Wang & Wright, 2016; Solingen et al., 2016a, b) and on their design loads (Kim et al., 2015). In the latter study, Kim et al. (2015) plots the spectrogram of the tower top signal obtained from nonlinear time simulations of a 2-bladed turbine and compares it to the spectrogram of a 3-bladed version of the same turbine. There are similarities between these spectrograms which lead the authors to conclude that 2-bladed turbines have similar modes as 3-bladed turbines. In experi-

mental study of a scaled turbine, Larsen & Kim (2015) conclude that asymmetric rotor modes split into backward and forward





whirling modes with $\pm 1$/rev, similar to whirling modes of 3-bladed turbines except that there also are components at multiple of the rotor speed.

In this paper, the modal dynamics of structures with rotor that have two and more blades is considered; first from a generic model-independent perspective, and then with focus on the differences between the modal dynamics of 2- and 3-bladed tur-

bines. In Section 2, analytical derivations of the linear equations of motion in a generic form and analytical inversion of the mass matrix show that the periodic system matrix for isotropic rotors with more than two blades have a finite and exact Fourier series with 3/rev being the highers harmonic order. The system matrix for structures with 2-bladed rotors has an infinite Fourier series of harmonic components that decrease in norm for increasing order. Using Hill's method to obtain the periodic mode shapes of the principal eigen-solutions, it is shown in Section 3 how the modal amplitudes for rotating blade degrees of free-

dom in the periodic eigenvectors can be used to decomposed the rotor motion into symmetric and anti-symmetric components for 2-bladed turbines and additional whirling components for rotors with more than two blades; noting that anti-symmetric components do not exist for odd number of blades and whirling components do not exist for 2-bladed turbines. In Section 4, a low-fidelity kinematic model of a 10MW turbine consisting of three blade modes and seven degrees of freedom for the nacelle and drivetrain is used to exemplify the differences between the modal dynamics of 2- and 3-bladed turbines using the presented

generic methods. It is shown that although a 2-bladed turbine do not have whirling modes, the response of an anti-symmetric rotor mode observed from a ground-fixed signal, such as a tower acceleration or moment, looks similar to the $\pm 1$/rev frequency splitting from the a whirling mode pair for a 3-bladed turbine. This similarity explains the incorrect conclusions made in the previous studies (Kim et al., 2015; Larsen & Kim, 2015). The present analytical study also shows that the additional harmonics observed for 2-bladed rotors lead to several significant modal couplings when the frequencies of higher harmonic components

in a periodic mode shape coincide with other modal frequencies.

## 2    Analytical system matrix for isotropic rotors

Analytical expressions for the harmonic components of the periodic system matrix for structures with an isotropic rotor are derived in this section. The first order state-space equation for a periodic system with the period $T$ is given by

$$\dot{\mathbf{x}} = \mathbf{A}(t)\mathbf{x} \tag{1}$$

where $\dot{(\,)} = d/dt$ and $\mathbf{A}(t) = \mathbf{A}(t+T)$ is the $T$-periodic system matrix of dimension $2N_D \times 2N_D$, where $N_D$ is the number of degrees of freedom (DOFs). Ordering the state vector as $\mathbf{x} = \{\mathbf{u}, \dot{\mathbf{u}}\}^T$ where $\mathbf{u}$ and $\dot{\mathbf{u}}$ are the DOFs and their time derivatives, the system matrix can be derived from linear second order equations of motion as

$$\mathbf{A}(t) = \begin{bmatrix} \mathbf{0} & \mathbf{I} \\ -\mathbf{M}^{-1}(t)\mathbf{K}(t) & -\mathbf{M}^{-1}(t)\mathbf{C}(t) \end{bmatrix} \tag{2}$$



where $\mathbf{I}$ is a identity matrix, and $\mathbf{M}$, $\mathbf{C}$, and $\mathbf{K}$ are the $T$-periodic mass, gyroscopic/damping, and stiffness $N_D \times N_D$ matrices, respectively. Periodicity of these matrices ensures that the system matrix can be written as a Fourier series

$$\mathbf{A}(t) = \sum_{n=-\infty}^{\infty} \mathbf{A}_n e^{in\Omega t} \qquad (3)$$

where $\imath = \sqrt{-1}$ and $\Omega \equiv 2\pi/T$ is the constant mean rotational speed. Note that the mean component of the system matrix $\mathbf{A}_0$ is a real matrix, and the complex matrices of the harmonic components come in conjugated pairs

$$\mathbf{A}_{-n} = \bar{\mathbf{A}}_n \, , \ \ n = 1, 2, 3, \dots \qquad (4)$$

where the bar denotes the complex conjugated operator. The periodic matrices of the second order equations of motion are derived in the next section. In Section 2.2, the mass matrix is inverted analytically to obtain its Fourier series. The mean and harmonic component matrices of the periodic system matrix are finally presented in Section 2.3.

## 2.1 Equations of motion

Let the Lagrangian for a structure with a rotor be written as

$$L = T(t, \mathbf{u}, \dot{\mathbf{u}}) - V(\mathbf{u}) \qquad (5)$$

where it is assumed that the potential energy of the conservative forces is time-independent and only depends on the generalized coordinates $V = V(\mathbf{u})$, e.g. the elastic forces. The total kinetic energy is given by an integral of the kinetic energy of each particle over the entire volume $\mathcal{V}$ of the structure as

$$T = \int_{\mathcal{V}} \tfrac{1}{2} \, \rho \, \dot{\mathbf{r}}^T \dot{\mathbf{r}} \, d\mathcal{V} \qquad (6)$$

where $(\cdot)^T$ denotes to the transpose matrix operator, and $\dot{\mathbf{r}}$ is the velocity vector of the particle given as the time derivative of its position vector $\mathbf{r} = \mathbf{r}(t, \mathbf{u})$, which for the rotor part of the structure will be explicitly time-dependent.

Substitution of the Lagrangian (5) with (6) into Lagrange's equations and linearization about a steady state deflection of the structure $\mathbf{u} = \mathbf{u}_0$ and $\dot{\mathbf{u}} = \mathbf{0}$, the coefficients of the matrices of Eq. (2) can be written as (Meirovitch, 1970)

$$
\begin{aligned}
m_{ij} &= \int_{\mathcal{V}} \rho \left( \frac{\partial \mathbf{r}^T}{\partial u_i} \frac{\partial \mathbf{r}}{\partial u_j} \right) d\mathcal{V} \\
c_{ij} &= \frac{\partial}{\partial t} m_{ij} + \int_{\mathcal{V}} \rho \left( \frac{\partial \mathbf{r}^T}{\partial u_i} \frac{\partial}{\partial u_j} \left( \frac{\partial \mathbf{r}}{\partial t} \right) - \frac{\partial \mathbf{r}^T}{\partial u_j} \frac{\partial}{\partial u_i} \left( \frac{\partial \mathbf{r}}{\partial t} \right) \right) d\mathcal{V} + \frac{\partial^2 D}{\partial \dot{u}_i \partial \dot{u}_j} \\
k_{ij} &= \int_{\mathcal{V}} \rho \frac{\partial}{\partial u_j} \left( \frac{\partial \mathbf{r}^T}{\partial u_i} \frac{\partial^2 \mathbf{r}}{\partial t^2} \right) d\mathcal{V} + \frac{\partial^2 V}{\partial u_i \partial u_j}
\end{aligned}
\qquad (7)
$$

where all derivatives are evaluated at $\mathbf{u} = \mathbf{u}_0$ and $D$ is Rayleigh's dissipation function. Note that only the partial time derivatives of the position vector is needed, not the full velocity, or acceleration vectors.





Let $\mathbf{u}_g$ denote the DOFs for the ground-fixed substructure and $\mathbf{u}_{b_k}$ denote the DOFs for rotating blade number $k$ (blades are in this paper always numbered in the direction of the rotation); then the position vector $\mathbf{r}$ in the ground-fixed inertial frame to a particle point on the substructure is written as a function of $\mathbf{u}_g$ as:

$$\mathbf{r} = \mathbf{r}_g(\mathbf{u}_g) \tag{8}$$

and on the blade number $k$ as

$$\mathbf{r} = \mathbf{r}_c(\mathbf{u}_g) + \mathbf{T}_c(\mathbf{u}_g) \left( \mathbf{R}_0 + \mathbf{R}_1 e^{\imath\psi_k} + \bar{\mathbf{R}}_1 e^{-\imath\psi_k} \right) \mathbf{r}_b(\mathbf{u}_{b_k}) \tag{9}$$

where the vector $\mathbf{r}_c$ and the rotation matrix $\mathbf{T}_c$ describe the position of the rotor center and the orientation of the rotational axis, respectively, both functions of the ground-fixed DOFs $\mathbf{u}_g$. The local position vector $\mathbf{r}_b$ of a particle on blade number $k$ is a function of $\mathbf{u}_{b_k}$, which is the same function for all blades due to the isotropy of the rotor and its discretization. The prescribed

rotation of the blade is given by the angle $\psi_k = \Omega t + 2\pi(k-1)/B$, where $B$ is the number of blades. The rotation matrix is written on exponential form using a real matrix $\mathbf{R}_0$ and a complex matrix $\mathbf{R}_1$, which are constant and given by the initial orientation of the rotational axis.

Let the conservative and dissipative forces be linear and depend only on the local DOFs and their time derivatives, such that the potential energy and Rayleigh's dissipation function can be written as

$$V = \mathbf{u}_g^T \mathbf{K}_g \mathbf{u}_g + \sum_{k=1}^{B} \mathbf{u}_{b_k}^T \mathbf{K}_b \mathbf{u}_{b_k} \text{ and } D = \dot{\mathbf{u}}_g^T \mathbf{C}_g \dot{\mathbf{u}}_g + \sum_{k=1}^{B} \dot{\mathbf{u}}_{b_k}^T \mathbf{C}_b \dot{\mathbf{u}}_{b_k} \tag{10}$$

where $\mathbf{C}_g$ and $\mathbf{K}_g$ are local damping and stiffness matrices for the ground-fixed substructure, and $\mathbf{C}_b$ and $\mathbf{K}_b$ are local and identical damping and stiffness matrices for each blade of the isotropic rotor.

Insertion of (10) and the position vectors (8) and (9) into the coefficients (7) and summation over all volumes of the entire structure, the linear equations of motion can be written on a generic matrix form as

$$\begin{bmatrix} \mathbf{M}_r & \mathbf{M}_{gr}^T \\ \mathbf{M}_{gr} & \mathbf{M}_g \end{bmatrix} \ddot{\mathbf{u}} + \begin{bmatrix} \mathbf{C}_r & \mathbf{G}_{rg} \\ \mathbf{G}_{gr} & \mathbf{G}_g + \mathbf{C}_g \end{bmatrix} \dot{\mathbf{u}} + \begin{bmatrix} \mathbf{K}_r + \mathbf{S}_r & \mathbf{0} \\ \mathbf{S}_{gr} & \mathbf{K}_g \end{bmatrix} \mathbf{u} = \mathbf{0} \tag{11}$$

where the $N_D$ DOFs are order as

$$\mathbf{u} = \{\mathbf{u}_r, \mathbf{u}_g\}^T \quad \text{with} \quad \mathbf{u}_r = \{\mathbf{u}_{b_1}, \dots, \mathbf{u}_{b_B}\}^T \tag{12}$$





Note that $N_D = N_g + BN_b$, where $N_g$ and $N_b$ are the number of DOFs on the substructure and each blade, respectively. The block matrices of (11) can be written as

$$\mathbf{M}_r = \text{diag}\{\mathbf{M}_b, \mathbf{M}_b, \cdots, \mathbf{M}_b\}, \quad \mathbf{C}_r = \text{diag}\{\mathbf{C}_b, \mathbf{C}_b, \cdots, \mathbf{C}_b\},$$

$$\mathbf{K}_r = \text{diag}\{\mathbf{K}_b, \mathbf{K}_b, \cdots, \mathbf{K}_b\}, \quad \mathbf{S}_r = \text{diag}\{\mathbf{S}_b, \mathbf{S}_b, \cdots, \mathbf{S}_b\},$$

$\quad\mathbf{M}_g(t) = \mathbf{M}_{g,0} + \mathbf{M}_{g,2}e^{\imath 2\Omega t} + \bar{\mathbf{M}}_{g,2}e^{-\imath 2\Omega t}$

$$\mathbf{G}_g(t) = \mathbf{G}_{g,0} + \mathbf{G}_{g,2}e^{\imath 2\Omega t} + \bar{\mathbf{G}}_{g,2}e^{-\imath 2\Omega t}$$

$$\mathbf{M}_{gr}(t) = \mathbf{M}_{gr,0} + \mathbf{M}_{gr,1}e^{\imath \Omega t} + \bar{\mathbf{M}}_{gr,1}e^{-\imath \Omega t},$$

$$\mathbf{G}_{gr}(t) = \mathbf{G}_{gr,0} + \mathbf{G}_{gr,1}e^{\imath \Omega t} + \bar{\mathbf{G}}_{gr,1}e^{-\imath \Omega t}, \tag{13}$$

$$\mathbf{G}_{rg}(t) = \mathbf{G}_{rg,1}e^{\imath \Omega t} + \bar{\mathbf{G}}_{rg,1}e^{-\imath \Omega t},$$

$\quad\mathbf{S}_{gr}(t) = \mathbf{S}_{gr,1}e^{\imath \Omega t} + \bar{\mathbf{S}}_{gr,1}e^{-\imath \Omega t}$

where the time-dependent coupling matrices can be subdivided further into constant single blade components as

$$\mathbf{M}_{gr,0} = [\mathbf{M}_{gb,0} \ \mathbf{M}_{gb,0} \ \cdots \ \mathbf{M}_{gb,0}],$$

$$\mathbf{M}_{gr,1} = \left[\mathbf{M}_{gb,1} \ \mathbf{M}_{gb,1}e^{i2\pi/B} \ \cdots \ \mathbf{M}_{gb,1}e^{i2\pi(B-1)/B}\right],$$

$$\mathbf{G}_{gr,0} = [\mathbf{G}_{gb,0} \ \mathbf{G}_{gb,0} \ \cdots \ \mathbf{G}_{gb,0}],$$

$\quad\mathbf{G}_{gr,1} = \left[\mathbf{G}_{gb,1} \ \mathbf{G}_{gb,1}e^{i2\pi/B} \ \cdots \ \mathbf{G}_{gb,1}e^{i2\pi(B-1)/B}\right], \tag{14}$

$$\mathbf{G}_{rg,1} = \left[\mathbf{G}_{bg,1}^T \ \mathbf{G}_{bg,1}^T e^{i2\pi/B} \ \cdots \ \mathbf{G}_{bg,1}^T e^{i2\pi(B-1)/B}\right]^T,$$

$$\mathbf{S}_{gr,1} = \left[\mathbf{S}_{gb,1} \ \mathbf{S}_{gb,1}e^{i2\pi/B} \ \cdots \ \mathbf{S}_{gb,1}e^{i2\pi(B-1)/B}\right]$$

The matrices in (13) and (14) related to inertia forces are listed in Appendix A. Note that the 2/rev components of the mass and gyroscopic matrices for the ground-fixed substructure only exists in case of a 2-bladed rotor.

**2.2 Inversion of mass matrix**

The inverse of the mass matrix in (11) can be written as

$$\mathbf{M}^{-1} = \begin{bmatrix} \mathbf{E} & \mathbf{F}^T \\ \mathbf{F} & \mathbf{H} \end{bmatrix} \tag{15}$$

where

$$\mathbf{E} = \mathbf{M}_r^{-1} + \mathbf{M}_r^{-1}\mathbf{M}_{gr}^T\mathbf{H}\mathbf{M}_{gr}\mathbf{M}_r^{-1}, \quad \mathbf{F} = -\mathbf{H}\mathbf{M}_{gr}\mathbf{M}_r^{-1},$$

$\quad\mathbf{H} = \left(\mathbf{M}_g - \mathbf{M}_{gr}\mathbf{M}_r^{-1}\mathbf{M}_{gr}^T\right)^{-1} \tag{16}$

Using (13) and (14), it can be shown that the inverse of $\mathbf{H}$ can be written as

$$\mathbf{H}^{-1} = \mathbf{Q}_0 + \mathbf{Q}_2 e^{\imath 2\Omega t} + \bar{\mathbf{Q}}_2 e^{-\imath 2\Omega t} \tag{17}$$





where the mean and 2/rev components for a $B$-bladed isotropic rotor are

$$\mathbf{Q}_0 = \mathbf{M}_{g,0} - B\left(\mathbf{M}_{gb,0}\mathbf{M}_b^{-1}\mathbf{M}_{gb,0}^T + \bar{\mathbf{M}}_{gb,1}\mathbf{M}_b^{-1}\mathbf{M}_{gb,1}^T + \mathbf{M}_{gb,1}\mathbf{M}_b^{-1}\bar{\mathbf{M}}_{gb,1}^T\right)$$

$$\mathbf{Q}_2 = \begin{cases} \mathbf{M}_{g,2} - 2\mathbf{M}_{gb,1}\mathbf{M}_b^{-1}\mathbf{M}_{gb,1}^T & \text{for } B = 2 \\ 0 & \text{for } B > 2 \end{cases} \tag{18}$$

Thus, $\mathbf{H} = \mathbf{Q}_0^{-1}$ is a real, symmetric, and constant matrix for isotropic rotors with more than two blades. Using (16) and (13), this property implies that the inverse mass matrix of such rotors have a finite Fourier series with 2/rev as the highest harmonic component. Note that the second harmonic component for 2-bladed rotors is also symmetric $\mathbf{Q}_2^T = \mathbf{Q}_2$,

For 2-bladed rotors, $\mathbf{H}^{-1}$ is periodic and $\mathbf{H}$ can therefore be written as a Fourier series

$$\mathbf{H} = \sum_{n=-\infty}^{\infty} \mathbf{H}_n e^{in\Omega t} \tag{19}$$

where $\mathbf{H}_{-n} = \bar{\mathbf{H}}_n$. Insertion into the equation $\mathbf{H}^{-1}\mathbf{H} = \mathbf{I}$ and collection of terms of equal harmonics yields

$$\mathbf{Q}_0\mathbf{H}_0 + \bar{\mathbf{Q}}_2\mathbf{H}_2 + \mathbf{Q}_2\bar{\mathbf{H}}_2 = \mathbf{I} \qquad \text{mean terms} \tag{20a}$$

$$\mathbf{Q}_0\mathbf{H}_1 + \bar{\mathbf{Q}}_2\mathbf{H}_3 + \mathbf{Q}_2\bar{\mathbf{H}}_1 = \mathbf{0} \qquad \text{1/rev terms} \tag{20b}$$

$$\mathbf{Q}_0\mathbf{H}_{2m+1} + \bar{\mathbf{Q}}_2\mathbf{H}_{2m+3} + \mathbf{Q}_2\mathbf{H}_{2m-1} = \mathbf{0} \qquad (2m\text{+}1)/\text{rev terms} \tag{20c}$$

$$\mathbf{Q}_0\mathbf{H}_{2m} + \bar{\mathbf{Q}}_2\mathbf{H}_{2m+2} + \mathbf{Q}_2\mathbf{H}_{2m-2} = \mathbf{0} \qquad (2m)/\text{rev terms} \tag{20d}$$

where $m = 1, 2, \ldots$ is a positive integer. The equations for odd terms are homogeneous and regular, thus all odd harmonic components vanish $\mathbf{H}_{2m-1} = \mathbf{0}$ for $m = 1, 2, \ldots$.

To solve the equations for the even terms, the mean component $\mathbf{H}_0$ is obtained from (20a) as a linear function of the second harmonic component $\mathbf{H}_2$ and a constant matrix

$$\mathbf{H}_0 = \mathbf{Q}_0^{-1} - \mathbf{Q}_0^{-1}\left(\bar{\mathbf{Q}}_2\mathbf{H}_2 + \mathbf{Q}_2\bar{\mathbf{H}}_2\right) \tag{21}$$

The remaining even equations can be solved recursively for $\mathbf{H}_{2m}$ by insertion of the solution for $\mathbf{H}_{2m-2}$ into the $2m$/rev equation. It is convenient to split the equations into real and imaginary parts and solve for each part to obtain

$$\begin{bmatrix} \text{Re}(\mathbf{H}_{2m}) \\ \text{Im}(\mathbf{H}_{2m}) \end{bmatrix} = \left(\prod_{k=1}^{m}\left(-\mathbf{P}_k\mathbf{Q}_s^T\right)\right)\mathbf{B}_0 - \mathbf{P}_m\mathbf{Q}_s\begin{bmatrix} \text{Re}(\mathbf{H}_{2m+2}) \\ \text{Im}(\mathbf{H}_{2m+2}) \end{bmatrix} \qquad \text{for} \quad m = 1, 2, \ldots \tag{22}$$

where the following real matrices have been introduced:

$$\mathbf{Q}_s = \begin{bmatrix} \text{Re}(\mathbf{Q}_2) & \text{Im}(\mathbf{Q}_2) \\ -\text{Im}(\mathbf{Q}_2) & \text{Re}(\mathbf{Q}_2) \end{bmatrix}, \quad \mathbf{B}_0 = \begin{bmatrix} \mathbf{Q}_0^{-1} \\ \mathbf{0} \end{bmatrix} \tag{23}$$

and the recursive matrices

$$\mathbf{P}_k = \left(\mathbf{I} - \mathbf{Q}_d^{-1}\mathbf{Q}_s^T\mathbf{P}_{k-1}\mathbf{Q}_s\right)^{-1}\mathbf{Q}_d^{-1} \quad \text{for} \quad k = 1, 2, \ldots \tag{24}$$



where $\mathbf{Q}_d = \mathrm{diag}\{\mathbf{Q}_0, \mathbf{Q}_0\}$ and $\mathbf{P}_0 = \mathrm{diag}\{2\mathbf{Q}_0^{-1}, \mathbf{0}\}$ are $2 \times 2$ block diagonal matrices. Symmetries of $\mathbf{Q}_0$ and $\mathbf{Q}_2$ and the resulting anti-symmetry of $\mathbf{Q}_s$ causes the matrices $\mathbf{P}_k$ to be symmetric. Note that $\mathbf{Q}_d$ is a regular matrix due to the positive definiteness of the mass matrix.

If $\left\| \mathbf{Q}_d^{-1} \mathbf{Q}_s^T \mathbf{P}_{k-1} \mathbf{Q}_s \right\| < 1$ then an inequality for the $p$-norm of the product $\mathbf{P}_k \mathbf{Q}_s^T$ can be derived (Golub & Van Loan, 1996) as

$$\left\| \mathbf{P}_k \mathbf{Q}_s^T \right\| \leq \frac{\left\| \mathbf{Q}_d^{-1} \mathbf{Q}_s^T \right\|}{1 - \left\| \mathbf{Q}_d^{-1} \mathbf{Q}_s^T \mathbf{P}_{k-1} \mathbf{Q}_s \right\|} \quad \text{for} \quad k = 1, 2, \ldots \tag{25}$$

where $\| \cdot \|$ here and subsequently denotes any $p$-norm of a matrix. The condition $\left\| \mathbf{Q}_d^{-1} \mathbf{Q}_s^T \mathbf{P}_{k-1} \mathbf{Q}_s \right\| < 1$ is fulfilled for $k \geq 1$ if $\left\| \mathbf{Q}_d^{-1} \mathbf{Q}_s^T \mathbf{P}_0 \mathbf{Q}_s \right\| < 1$. It is not straight forward to prove this inequality, but it's validity for a given model can easily be checked. Intuitively, the $p$-norm of the constant part $\mathbf{Q}_0$ of the mass matrix for the ground-fixed coordinates should be much larger then the $p$-norm of the second order harmonic part $\mathbf{Q}_2$. It is therefore also assumed that $\left\| \mathbf{P}_k \mathbf{Q}_s^T \right\| < 1$ for $k = 1, 2, \ldots$ based on the inequality (25). From the recursive solution (22), this assumption is sufficient (but not necessary) to ensure that the $p$-norm of harmonic components $\mathbf{H}_{2m}$ decreases with their order

$$\| \mathbf{H}_{2m} \| \leq \left( \prod_{k=1}^{m} \left\| \mathbf{P}_k \mathbf{Q}_s^T \right\| \right) \| \mathbf{B}_0 \| + \| \mathbf{P}_m \mathbf{Q}_s \| \, \| \mathbf{H}_{2m+2} \| \to 0 \quad \text{for} \quad m \to \infty \tag{26}$$

Thus, closure to the recursive equation (22) can therefore be obtained by choosing $\mathbf{H}_{2N_H+2} = \mathbf{0}$, where $2N_H$ is the highest harmonic component used in the Fourier series (19). The recursive solution of (22) is computed backward starting with $\mathbf{H}_{2N_H}$ and ending with $\mathbf{H}_0$ given by Eq. (21).

Truncation of (19) to $2N_H$ and insertion into (16) shows that the block matrices of the inverse mass matrix are

$$\mathbf{E} = \sum_{n=-(2N_h+2)}^{2N_h+2} \mathbf{E}_n e^{\imath n \Omega t}, \quad \mathbf{F} = \sum_{n=-(2N_h+1)}^{2N_h+1} \mathbf{F}_n e^{\imath n \Omega t}, \quad \text{and} \quad \mathbf{H} = \sum_{m=-N_h}^{N_h} \mathbf{H}_{2m} e^{\imath 2m \Omega t} \tag{27}$$

where the component matrices $\mathbf{E}_n$ and $\mathbf{F}_n$ are written out in Appendix B. Thus, the highest order of the harmonics in the inverse mass matrix for a 2-bladed rotor is $2N_H + 2$ and involves only the rotor coordinates.

## 2.3 Harmonic components in system matrix

Insertion of (15) with (27) into the system matrix (2) with the gyroscopic/damping and stiffness matrices of (11) shows that the Fourier series of the periodic system matrix (3) can be truncated to the order $N$ as

$$\mathbf{A}(t) = \sum_{n=-N}^{N} \mathbf{A}_n e^{\imath n \Omega t} \tag{28}$$

where

$$N = \begin{cases} 2N_H + 3 & \text{for } B = 2 \\ 3 & \text{for } B > 2 \end{cases} \tag{29}$$



and the matrices $\mathbf{A}_n$ can be found in Appendix C. This analytical derivation of the system matrix is exact for isotropic rotors with mode than two blades. In fact, if the anisotropy of the rotor is only related to the stiffnesses of the blades and not the mass distributions or rotor geometry, as in previous studies e.g. (Skjoldan & Hansen, 2009; Skjoldan, 2009), then the system matrices for such rotors also have finite Fourier series. For 2-bladed isotropic rotors, the series converges when the Fourier

series of the inverse mass matrix converges; sufficient but not necessary criteria for convergence are given in the previous section.

## 3 Modal analysis using Hill's method

This section contains a description of Hill's method and how it can be applied for modal analysis of structures with bladed rotors. First, the concept of periodic mode shapes and Hill's truncated eigenvalue problem is introduced. Then, a novel method

for automatic identification of the principal solutions among all eigen-solutions of this eigenvalue problem is briefly presented. The section ends with a description of the modes of bladed rotors, including the identification of the different rotor mode components based on the periodic eigenvectors of Hill's eigenvalue problem.

### 3.1 Periodic mode shapes and Hill's truncated eigenvalue problem

Floquet theory defines that the eigen-solution of the linear periodic system (1) consists of an eigenvalue and a corresponding

periodic eigenvector. A homogenous solution to (1) can therefore be written as

$$\mathbf{x} = \sum_{m=-\infty}^{\infty} \mathbf{v}_m e^{\imath m \Omega t} e^{\lambda t} \tag{30}$$

where $\mathbf{v}_m$ are harmonic components of the periodic eigenvector and $\lambda$ is the complex eigenvalue. Insertion into (1) with the $2N_d \times 2N_D$ periodic system matrix written as the infinite Fourier series (3) and collecting terms of equal harmonics yields an infinite set of $2N_D$ equations

$$\sum_{n=-\infty}^{\infty} \mathbf{A}_n \mathbf{v}_{n-m} - (\lambda + \imath m \Omega) \mathbf{v}_m = \mathbf{0} \ \forall m \in \mathbb{Z} \tag{31}$$

These equations constitutes the algebraic eigenvalue problem of infinite dimension that forms the basis of Hill's method (Hill, 1886; Xu & Gasch, 1995). The eigenvalue is $\lambda$ and the eigenvector can written as $\mathbf{v} = \{\ldots, \mathbf{v}_{-2}^T, \mathbf{v}_{-1}^T, \mathbf{v}_0^T, \mathbf{v}_1^T, \mathbf{v}_2^T, \ldots\}^T$ and is of infinite dimension. If the harmonic index in the eigenvector is shifted by an integer $s$ then the resulting vector $\mathbf{v} = \{\ldots, \mathbf{v}_{-2-s}^T, \mathbf{v}_{-1-s}^T, \mathbf{v}_{-s}^T, \mathbf{v}_{1-s}^T, \mathbf{v}_{2-s}^T, \ldots\}^T$ and the complex number $\lambda + \imath s \Omega$ are also an eigen-solution. Thus, although the

eigenvalue problem has infinitely many eigen-solutions, there are only $2N_D$ principal solutions from which all other solutions can be constructed. This construction is also clear when inserting the shifted eigen-solution into (30)

$$\mathbf{x} = \sum_{m=-\infty}^{\infty} \mathbf{v}_{m-s} e^{\imath m \Omega t} e^{(\lambda + \imath s \Omega)t} = \sum_{m=-\infty}^{\infty} \mathbf{v}_m e^{\imath (m-s)\Omega t} e^{(\lambda + \imath s \Omega)t} = \sum_{m=-\infty}^{\infty} \mathbf{v}_m e^{\imath m \Omega t} e^{\lambda t} \tag{32}$$

This arbitrary $s$/rev shift between the periodicity of the eigenvector and the frequency of the eigenvalue is identical to the frequency indeterminacy in Floquet analysis due to the logarithm of the complex Floquet multipliers (Skjoldan & Hansen, 2009;





Bottasso & Cacciola, 2015). The advantage of Floquet analysis is that there is only $2N_D$ solutions and the frequency indeterminacy can be chosen arbitrarily for each of them; here the concept of *participation factors* introduced by Bottasso & Cacciola (2015) is helpful. In Hill's method, the problem is to link all eigen-solutions into $2N_D$ sub-sets in which a principal eigen-solution can be used to construct all eigen-solutions in the particular sub-set. A novel method for this linking into sub-sets and identification of the principal solutions to Hill's eigenvalue problem is presented in the next section.

To numerically solve Hill's eigenvalue problem (31) is truncated. When the periodic system matrix has a finite Fourier series (28), the periodic eigenvectors can also be represented by a finite Fourier series (Curtis, 2010) in the homogenous solution

$$\mathbf{x} = \sum_{m=-M}^{M} \mathbf{v}_m e^{\imath m \Omega t} e^{\lambda t} \tag{33}$$

where $M$ is the number of harmonics. The infinite eigenvalue problem (31) can then be truncated to finite dimension as

$$\sum_{n=-N_m}^{N_m} \mathbf{A}_n \mathbf{v}_{n-m} - (\lambda + \imath m \Omega) \mathbf{v}_m = \mathbf{0} \quad \forall m \in [-M:M] \tag{34}$$

where $N_m = \min(N, |M - m|)$ is the limit for the summation over the product of the harmonic components of the system matrix and the eigenvector. This limit is lower than the number of harmonics in the system matrix $N$ for the matrix equations where $|m| > M - N$, showing that truncation errors are introduced in those $2N$ of the $2M + 1$ matrix equations in Eq. (34). The truncation error has been investigated by Skjoldan (2009); Lee et al. (2007) and they show a convergence of the principal eigen-solutions of the truncated Hill's matrix for their particular systems when $M \geq 2N$. This finite matrix can be easily set up from (34) by arranging the harmonic components of the periodic eigenvector as

$$\mathbf{v} = \{\mathbf{v}_{-M}^T, \dots, \mathbf{v}_{-1}^T, \mathbf{v}_0^T, \mathbf{v}_1^T, \dots, \mathbf{v}_M^T\}^T \tag{35}$$

The $2N_D(2M+1)$ solutions to the truncated eigenvalue problem (34) will still follow the above rules for index shift, except that the eigen-solutions with the largest harmonic components of their eigenvectors closest to the "edges" at $\pm M$ will be affected by the truncation error. There still exist $2N_D$ sub-sets of $2M + 1$ eigen-solutions that can be constructed from a principal solution; however, these solutions with periodic eigenvectors of highest harmonic order will be less accurately constructed. Note that the eigenvector (35) can be given a norm of 1, such that $\|\mathbf{v}_m\|$ is the participation factor of the $m$'th harmonic component (Bottasso & Cacciola, 2015).

## 3.2 Automatic identification of principal solutions

The identification of the principal solutions can for small systems be done manually (Skjoldan, 2009; Christensen & Santos, 2005; Lee et al., 2007; Kim & Lee, 2012). More systematic approach is to select the $2N_D$ eigen-solutions among the $2N_D(2M + 1)$ solutions that have periodic eigenvectors where the harmonic components are most centered around the mean component, different methods of this concept can be found in (Xu & Gasch, 1995; Ertz et al., 1995; Lazarus & Thomas, 2010). However, these methods do not always ensure that the identified principal solutions $2N_D$ can construct all $2N_D(2M + 1)$ solutions,



because two selected principal solutions may come from the same sub-set, whereby one sub-set is not represented. A novel method is therefore suggested, where the principal solutions are identified in three steps:

1. Remove half the eigen-solutions with eigenvectors dominated by harmonic components of the highest order.

2. Link the remaining eigen-solutions in $2N_D$ sub-sets based on a Modal Assurance Criterion.

3. Pick the principal solution in each sub-set that has the largest mean component $\mathbf{v}_0$ in its eigenvector.

Step 1 ensures that the eigen-solutions with the largest truncation errors are removed. Step 2 ensures that all $2N_D$ sub-sets are represented. The choice of the particular solution in step 3 as the principal one in each sub-set is less important (similar to Floquet analysis). If the eigenvector with the largest norm of its mean ground-fixed components $||\mathbf{v}_{0,g}||$ is chosen, then the frequency of the principal eigenvalue will also be the dominating frequency observed in the ground-fixed frame

(Skjoldan & Hansen, 2009). For drawing of Campbell diagrams, it is convenient to link the sub-sets across the variation of rotor speed. Step 3 is therefore only done for one rotor speed computation, and the selection of the principal solutions in each sub-set for subsequent rotor speeds is based on a Modal Assurance Criterion with the previous speed.

### 3.3   Modes of $B$-bladed isotropic rotors

Modes of a structure with a bladed rotor may be dominated by the motion of the sub-structure and therefore named after its

dominant component of the periodic eigenvector, e.g. 'tower fore-aft' or 'drivetrain torsion' modes of wind turbines. The name of a rotor mode dominated by blade motion will depend on the number of blades.

The naming conventions of symmetric and whirling rotor modes of 3-bladed rotors deduced from the modal analysis using the multi-blade coordinates (Hansen, 2003) can be generalized. The Coleman transformation will for isotropic rotors with more than two blades render the system matrix time-invariant with constant eigenvectors described in multi-blade coordinates

(Skjoldan & Hansen, 2009). The Coleman transformation can be written on a complex form, where the cosine and sine parts of the multi-blade coordinates are combined into backward and forward whirling coordinates, similar to complex wave coordinates for vibrations of spinning disks (Lee & Kim, 1995; Hansen, 1999). Let the modal response in these coordinates be written as $\mathbf{z}_r e^{\imath\omega t}$ then the Coleman transformed responses in rotating blade coordinates will be given as

$$
\mathbf{u}_r = \begin{bmatrix}
\mathbf{I} & -\mathbf{I} & \mathbf{I}e^{\imath\psi_1} & \mathbf{I}e^{-\imath\psi_1} & \cdots & \mathbf{I}e^{\imath\tilde{B}\psi_1} & \mathbf{I}e^{-\imath\tilde{B}\psi_1} \\
\mathbf{I} & \mathbf{I} & \mathbf{I}e^{\imath\psi_2} & \mathbf{I}e^{-\imath\psi_2} & \cdots & \mathbf{I}e^{\imath\tilde{B}\psi_2} & \mathbf{I}e^{-\imath\tilde{B}\psi_2} \\
\mathbf{I} & -\mathbf{I} & \mathbf{I}e^{\imath\psi_3} & \mathbf{I}e^{-\imath\psi_3} & \cdots & \mathbf{I}e^{\imath\tilde{B}\psi_3} & \mathbf{I}e^{-\imath\tilde{B}\psi_3} \\
\vdots & \vdots & \vdots & \vdots & & \vdots & \vdots \\
\mathbf{I} & \mathbf{I} & \mathbf{I}e^{\imath\psi_B} & \mathbf{I}e^{-\imath\psi_B} & \cdots & \mathbf{I}e^{\imath\tilde{B}\psi_B} & \mathbf{I}e^{-\imath\tilde{B}\psi_B}
\end{bmatrix} \mathbf{z}_r e^{\imath\omega t} \tag{36}
$$

where $\tilde{B} = (B-1)/2$ for odd and $\tilde{B} = B/2-1$ for even number of blades, $\psi_k = \Omega t + 2\pi(k-1)/B$ is the azimuth angle of blade number $k$, the second column in the matrix is only present for even number of blades, and the harmonic azimuth dependent parts are omitted for 2-bladed rotors. These harmonic parts come in pairs for each harmonic order with plus and minus on





the blade azimuth angle, defining the direction of the whirling. The constant eigenvector in complex whirling coordinates can be written as $\mathbf{z}_r = \{\mathbf{A}_0^T, \mathbf{A}_D^T, \mathbf{A}_{BW,1}^T, \mathbf{A}_{FW,1}^T, \dots, \mathbf{A}_{BW,\tilde{B}}^T, \mathbf{A}_{FW,\tilde{B}}^T\}^T$, which by substitution into (36) shows that the modal response of blade number $k$ of the isotropic rotor can be written as

$$\mathbf{u}_{b_k} = \left(\mathbf{A}_0 + (-\mathbf{I})^k \mathbf{A}_D\right) e^{\imath \omega t} + \sum_{p=1}^{\tilde{B}} \mathbf{A}_{\mathrm{BW},p} \, e^{\imath(\omega+p\Omega)t} e^{\imath \frac{2\pi p(k-1)}{B}} + \sum_{p=1}^{\tilde{B}} \mathbf{A}_{\mathrm{FW},p} \, e^{\imath(\omega-p\Omega)t} e^{-\imath \frac{2\pi p(k-1)}{B}} \tag{37}$$

showing that $\mathbf{A}_0$ are symmetric components, $\mathbf{A}_D$ are anti-symmetric components, and $\mathbf{A}_{\mathrm{BW},p}$ and $\mathbf{A}_{\mathrm{FW},p}$ are backward (BW) and forward whirling (FW) components of the blade motion in the mode. The direction of the whirl is given by the sign of the phase shifts $\frac{2\pi p}{B}(k-1)$ for each blade. Note that for BW components, the angular frequency $p$/rev is added to the eigenfrequency $\omega$, and it is subtracted for FW modes. As explained in Hansen (2003, 2007), the eigenfrequencies of a 3-bladed rotor system described in multi-blade coordinates are measured in the ground-fixed frame, in which the frequencies of a pure

BW mode $\omega_{BW}$ and pure FW mode $\omega_{FW}$ decrease and increase with 1/rev, respectively, such that their frequencies in the rotating blade frame given by (37) are close to the frequency of the corresponding blade mode $\omega_b \approx \omega_{BW} + \Omega \approx \omega_{FW} - \Omega$. Equation (37) shows that generally there will be whirling mode pairs that split with up to $2\tilde{B}$/rev in the ground-fixed frame, where the general relationship $\omega_b \approx \omega_{BW} + p\Omega \approx \omega_{FW} - p\Omega$ to the blade frequencies holds. Note that the phase speed $\frac{2\pi p}{B}$ of the rotor whirling will a factor $p$ higher. Note also that the frequencies of anti-symmetric components for even number of blades

(except for two) are unchanged by the transformation from ground-fixed to rotating blade frame. This properties shared by the symmetric components is caused be the fact that the reaction forces due to these components of rotor motion is neutral with respect to the rotation; the center of gravity of the entire rotor is only moved axially by symmetric components and remains stationary for anti-symmetric components.

Each harmonic component of each DOF of a rotor mode may therefore be named

– Symmetric

– Anti-symmetric (exist only for $B$ even)

– Backward whirling (exist only for $B \geq 3$)

– Forward whirling (exist only for $B \geq 3$)

followed by the name of the DOF. For rotors with more than four blades, the whirling components must also have the phase

speed index $p = 1, 2, \dots \tilde{B}$ added to the name of the particular component. The symmetric, anti-symmetric, and whirling components of the rotor motion given by (37) can be derived by inverting the transformation (36). Inserting the rotor motion $\mathbf{u}_r = \mathbf{v}_{m,r} e^{\imath m\Omega t} e^{\lambda t}$ of the $m$'th harmonic of the periodic eigenvector into (36) and setting the time to zero $t = 0$, the symmet-





ric, anti-symmetric, and whirling components of this harmonic can be derived as

$$
\begin{Bmatrix}
\mathbf{A}_0 \\
\mathbf{A}_D \\
\mathbf{A}_{BW,1} \\
\mathbf{A}_{FW,1} \\
\vdots \\
\mathbf{A}_{BW,\tilde{B}} \\
\mathbf{A}_{FW,\tilde{B}}
\end{Bmatrix}_m
= \frac{1}{B}
\begin{bmatrix}
\mathbf{I} & \mathbf{I} & \dots & \mathbf{I} & \mathbf{I} \\
-\mathbf{I} & \mathbf{I} & \dots & -\mathbf{I} & \mathbf{I} \\
\mathbf{I} & \mathbf{I}e^{-\imath 2\pi/B} & \dots & \mathbf{I}e^{-\imath 2\pi(B-1)/B} & \mathbf{I} \\
\mathbf{I} & \mathbf{I}e^{\imath 2\pi/B} & \dots & \mathbf{I}e^{\imath 2\pi(B-1)/B} & \mathbf{I} \\
\vdots & \vdots & & \vdots & \vdots \\
\mathbf{I} & \mathbf{I}e^{-\imath \tilde{B}2\pi/B} & \dots & \mathbf{I}e^{-\imath \tilde{B}2\pi(B-1)/B} & \mathbf{I} \\
\mathbf{I} & \mathbf{I}e^{\imath \tilde{B}2\pi/B} & \dots & \mathbf{I}e^{\imath \tilde{B}2\pi(B-1)/B} & \mathbf{I}
\end{bmatrix}
\mathbf{v}_{m,r}
\tag{38}
$$

where $\mathbf{v}_{m,r}$ is the rotor part of a harmonic component $\mathbf{v}_m$ of the eigenvector $\mathbf{v}$, and the last row of matrices is omitted for rotors with odd number of blades. The harmonic order $m$ of each component is also relevant for its naming, but again noting that it is directly dependent on the choice of the principal solution. Note that 2-bladed rotors only have symmetric and anti-symmetric components.

## 4   Modal analysis of 2- and 3-bladed wind turbines

The theories presented in the previous sections are applicable to structures with isotropic rotors with any number of blades higher than one. In this section, the modal dynamics of 2- and 3-bladed turbines are investigated because they are of the highest interest to the wind turbine industry, but also because the finite Fourier series of the system matrix shows that there are no qualitative difference between turbines having three or having more identical blades.

The turbine used for the analysis is the DTU 10MW reference wind turbine (RWT) by Bak et al. (2012) with three identical blades. For the sake of comparability of the modal dynamics, the 2-bladed version is obtained by reusing the same blade. In reality, the optimal aerodynamic solidity of the rotor would require a redesign (Bergami et al., 2014); the blades would get a larger chord and the increased absolute thickness (assuming the same relative thickness of the airfoils) would be used to either decrease the blade mass, increase the blade stiffness (keeping the same bending stresses), or combine these two objectives. In any cases, the blade for the 2-bladed turbine will have different blade modal frequencies and possible mass distribution, which would complicate the direct comparison of turbine modes to the 3-bladed version.

The turbine model derived in the next section is based on the simple model presented in Hansen (2003), except that the bending of the main shaft is here omitted and the generator rotation has been added as a new DOF. The model parameters are tuned such that the modal frequencies of the first eleven modes of the three-bladed turbine are closest possible to the modal frequencies computed for the 10MW RWT with the higher fidelity linear model of the software HAWCStab2 using beam elements and the method of the Coleman transformation (Hansen, 2004). Minor modifications of the turbine have been introduced such that the center of gravity in the blade cross-sections coincide with the pitch axis and the coning is also set to zero.

The convergence of the Fourier series of the system matrix for the 2-bladed turbine is analyzed in Section 4.2. The Campbell diagrams of the principal modal frequencies are presented for 3- and 2-bladed turbines in Section 4.3. The well-known periodic





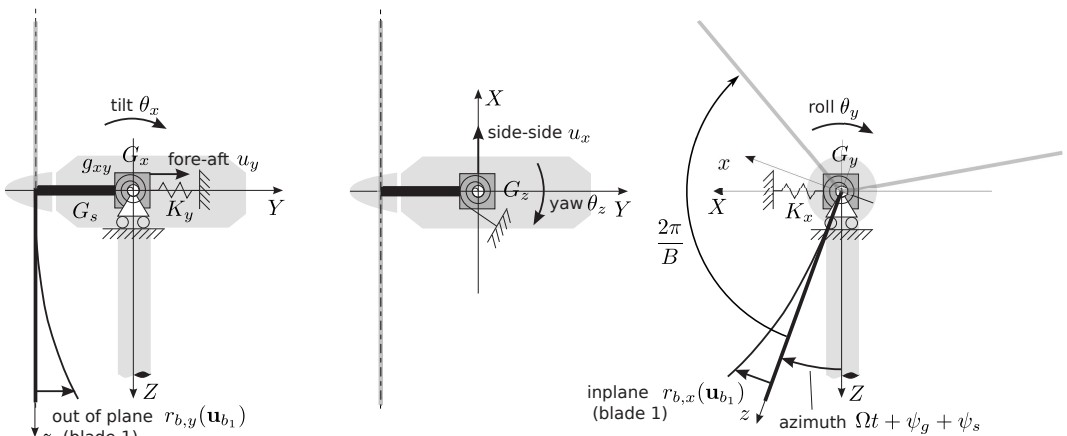

**Figure 1.** Illustration of the simple turbine model.

mode shapes of 3-bladed turbines are repeated in Section 4.4, and more complex periodic mode shapes for 2-bladed turbines are presented in Section 4.5. The section ends with a discussion of the differences between the modal dynamics of the two turbine types.

### 4.1 Model kinematics and parameters

Figure 1 shows an illustration of the structural turbine model. The nacelle and tower motions are described by five DOFs. The nacelle can translate in the two horizontal directions described in the ground-fixed inertial frame $(X, Y, Z)$ by $u_x$ (side-side) and $u_y$ (fore-aft). It can tilt, roll and yaw described by the angles $\theta_x$, $\theta_y$, and $\theta_z$, respectively. The azimuthal angle of the blade number one is $\psi_1 = \Omega t + \psi_s + \psi_g$, where $\Omega$ is the constant mean speed, and $\psi_s$ and $\psi_g$ are the torsional and rigid-body rotations of the drivetrain, respectively. The generator is rotating at the speed $\Omega + \dot{\psi}_g$.

The blade motion is described in their own rotating frames $(x, y, z)$, where the $z$-axis is the blade axis and the $y$-axis at rest coincides with the $Y$-axis. The local position vector for the center of gravity on blade number $k$ is described by an expansion in the first three blade modes at standstill as

$$\mathbf{r}_b(\mathbf{u}_{b_k}) = \begin{Bmatrix} \phi_{x,f_1}(z) \\ \phi_{y,f_1}(z) \\ z \end{Bmatrix} q_{f_1,k}(t) + \begin{Bmatrix} \phi_{x,e}(z) \\ \phi_{y,e}(z) \\ z \end{Bmatrix} q_{e,k}(t) + \begin{Bmatrix} \phi_{x,f_2}(z) \\ \phi_{y,f_2}(z) \\ z \end{Bmatrix} q_{f_2,k}(t) \tag{39}$$

for $z \in [R_h : R]$ and $\mathbf{r}_b = \{0, 0, z\}^T$ for $z \in [0 : R_h]$, where $R_h$ and $L_b$ are the hub radius and blade length, respectively. The

outer rotor radius is $R = R_h + L_b$. The edgewise and flapwise deflections in the first flapwise blade mode are $\phi_{x,f_1}(z)$ and $\phi_{y,f_1}(z)$, respectively, and $q_{f_1,k}$ is the DOF of this deflection shape for blade number $k$. Similar, the subscripts $e$ and $f_2$ denote the contributions from the first edgewise and second flapwise deflection shapes. All shape functions are obtained by polynomial fits to the isolated blade mode shapes computed with the beam element model of HAWCStab2, see Figure 2.





The vector containing the system DOFs is defined according to Eq. (12) as

$$\mathbf{u} = \{q_{f_1,1}, q_{e,1}, q_{f_2,1}, \ldots, q_{f_1,B}, q_{e,B}, q_{f_2,B}, u_x, u_y, \theta_x, \theta_y, \theta_z, \psi_g, \psi_s\}^T \tag{40}$$

where the number of DOFs is dependent on the number of blades as $N_D = 3B + 7$. To obtain the linear equations of motion using the derivations of Section 2.1, the blade mass motion is written on the form of (9) using (39) and the following rotor

center position and orientation of the rotational axis

$$\mathbf{r}_c = \begin{bmatrix} 1 & -\theta_z & \theta_y \\ \theta_z & 1 & -\theta_x \\ -\theta_y & \theta_x & 1 \end{bmatrix} \begin{Bmatrix} 0 \\ -L_s \\ 0 \end{Bmatrix}$$

$$\mathbf{T}_c = \begin{bmatrix} 1 & -\theta_z & \theta_y \\ \theta_z & 1 & -\theta_x \\ -\theta_y & \theta_x & 1 \end{bmatrix} \begin{bmatrix} \cos(\psi_s + \psi_g) & 0 & \sin(\psi_s + \psi_g) \\ 0 & 1 & 0 \\ -\sin(\psi_s + \psi_g) & 0 & \cos(\psi_s + \psi_g) \end{bmatrix} \tag{41}$$

and constant rotation matrices

$$\mathbf{R}_0 = \begin{bmatrix} 1 & 0 & 0 \\ 0 & 0 & 0 \\ 0 & 0 & 1 \end{bmatrix} \quad \text{and} \quad \mathbf{R}_1 = \begin{bmatrix} 1/2 & 0 & -\imath/2 \\ 0 & 0 & 0 \\ \imath/2 & 0 & 1/2 \end{bmatrix} \tag{42}$$

These vector and matrix functions of the DOFs are inserted into the volume integrals for the matrix elements in Appendix A, which reduce to line integrals over $z \in [0 : R]$. The mass distribution of hub is defined as $m(z) = M_h/B/R_h$ for $z \in [0 : R_h]$, where $M_h$ is the total hub mass. The mass distribution of the blade is plotted in Figure 2.

The ground-fixed substructure are modeled as a lumped mass, and the inertia forces from the nacelle and effective tower masses and the generator rotational inertia are derived from the kinetic energy:

$T_g = \frac{1}{2}M(\dot{u}_x + \dot{u}_y)^2 + \frac{1}{2}I_x\dot{\theta}_x^2 + \frac{1}{2}I_y\dot{\theta}_y^2 + \frac{1}{2}I_z\dot{\theta}_z^2 + \frac{1}{2}I_g\dot{\psi}_g^2$  (43)

such that the first term of $m_{g,0,ij}$ in Eq. (A2) is replaced by $\partial^2 T_g / \partial \dot{u}_i \partial \dot{u}_j$. The potential energy for the linear elastic stiffnesses of the nacelle/tower motion, shaft torsion and blade deflections is written as

$$V = \frac{1}{2}K_x u_x^2 + \frac{1}{2}K_y u_y^2 + \frac{1}{2}G_x\theta_x^2 + \frac{1}{2}G_y\theta_y^2 + \frac{1}{2}G_z\theta_z^2 - g_{xy}\theta_x u_y + g_{xy}\theta_y u_x + \frac{1}{2}G_s\psi_s^2 + \sum_{k=1}^{B} V_{b_k} \tag{44}$$

where $V_{b_k}$ is the potential energy of blade number $k$ given as

$$V_{b_k} = \sum_{\beta=[f_1,e,f_2]} \frac{1}{2}M_\beta\omega_\beta^2 q_{\beta,k}^2 + \Omega^2 \sum_{\beta=[f_1,e,f_2]} \int_0^R \left( \left(\frac{\partial^2\phi_{x,\beta}}{\partial z^2}\right)^2 + \left(\frac{\partial^2\phi_{y,\beta}}{\partial z^2}\right)^2 \right) q_{\beta,k}^2 \int_z^R m(s)\,ds\,dz \tag{45}$$

where $M_\beta$ and $\omega_\beta$ are the modal mass and frequency of the orthogonal blade modes $\beta = [f_1, e, f_2]$. The last term is the potential energy of the centrifugal forces proportional to the squared rotor speed.




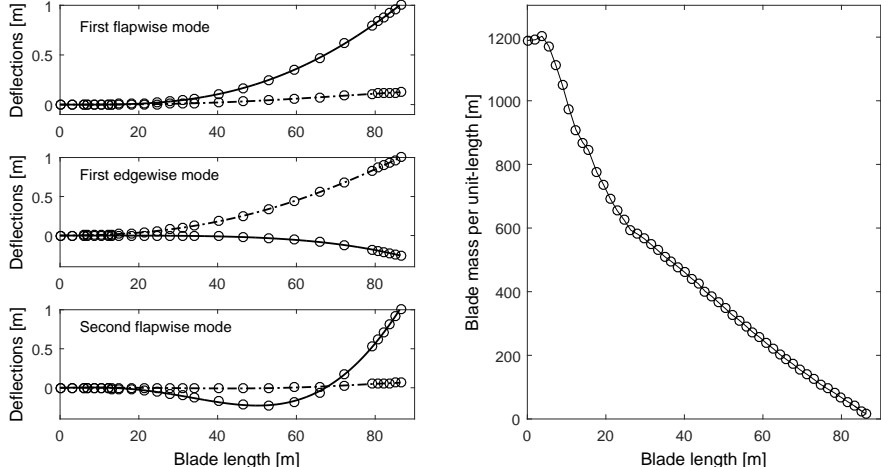

**Figure 2.** Edgewise (dashed curves) and flapwise (solid curves) deflections in the blade mode shapes (left plots) and the blade mass distribution (right plot) for the blade of the DTU 10MW RWT. Circles in the left plots are results from the beam element model of the software HAWCStab2 and the curves are polynomial fits used in the present model.

The damping matrices of Rayleigh's dissipation function in (10) for the blades $\mathbf{C}_b$ and the nacelle/tower motion and shaft torsion $\mathbf{C}_g$ are setup using a spectral damping model (Clough & Penzien, 1975). The first and second flapwise blade modes are set to have logarithmic decrements of 20 % and 10 %, respectively, whereas all other modes are damped 2 – 5 %. These choices of damping are simply crude approximations to the aeroelastic damping of the turbine in operation (Bak et al., 2012), and they

have no significant effect on the results of Hill's method. Damping is very important for solutions using Floquet theory where the system is time integrated.

The linear equations of motion (11) can be derived from Eq. (7) and the integrals in Appendix A using the above kinematic description of the blade motion, the kinetic energy of the nacelle/tower and generator inertia, the total potential energy, and the spectral damping matrices. The block matrices of (11) are not explicitly included for brevity. All parameters (except for the

blade properties in Figure 2) of the tuned model are listed in Table 1.

### 4.2 Convergence of Fourier series for 2-bladed rotor

Convergence of the Fourier series for the system matrix of 2-bladed isotropic rotors is ensured if the constant part of the mass matrix for the ground-fixed DOFs is sufficiently larger then the second order harmonic part. Using Eq. (23) and (24), it can be computed that $\left\|\mathbf{Q}_d^{-1}\mathbf{Q}_s^T\mathbf{P}_0\mathbf{Q}_s\right\| \approx 0.22 < 1$ and $\left\|\mathbf{P}_k\mathbf{Q}_s^T\right\| < 0.67$ for $k = 1, 2, \ldots$, i.e., the inverse mass matrix and therefore

the system matrix have finite Fourier series for the 2-bladed rotor.

Figure 3 shows the 2-norms of the Fourier components of the system matrices for the models with the 2- and 3-bladed isotropic rotors. The convergence for the 2-bladed rotor is observed and for the following analysis the number of harmonics is chosen to be $N = 7$. The largest norm for the 1/rev component is 46,480 s$^{-1}$ and the largest norm of an omitted component



| Parameter description | Symbol | Value |
|---|---|---|
| Blade length | $L_b$ | 86.366 m |
| First blade flap frequency | $\omega_{f_1}$ | 0.610 Hz |
| First blade edge frequency | $\omega_e$ | 0.934 Hz |
| Second blade flap frequency | $\omega_{f_2}$ | 1.738 Hz |
| Hub radius | $R_h$ | 2.8 m |
| Hub mass | $M_h$ | 105,520 kg |
| Tower to rotor distance | $L_s$ | 7.1 m |
| Generator inertia on LSS | $I_g$ | 3,751,000 kgm$^2$ |
| Drivetrain stiffness | $G_s$ | 0.668 GNm/rad |
| Nacelle/effective tower mass | $M$ | 446,040 kg |
| Nacelle tilt inertia | $I_x$ | 4,106,000 kgm$^2$ |
| Nacelle roll inertia | $I_y$ | 410,600 kgm$^2$ |
| Nacelle yaw inertia | $I_z$ | 4,106,000 kgm$^2$ |
| Tower top side-side stiffness | $K_x$ | 7.4 MN/m |
| Tower top fore-aft stiffness | $K_y$ | 7.4 MN/m |
| Tower top tilt stiffness | $G_x$ | 7.462 GNm/rad |
| Tower top roll stiffness | $G_y$ | 7.462 GNm/rad |
| Tower top coupling stiffness | $g_{xy}$ | 0.2035 GN |
| Tower top yaw stiffness | $G_z$ | 3.5 GNm/rad |

**Table 1.** Tuned parameters of simple model to fit the modal properties of the DTU 10MW RWT up till its 11th mode.

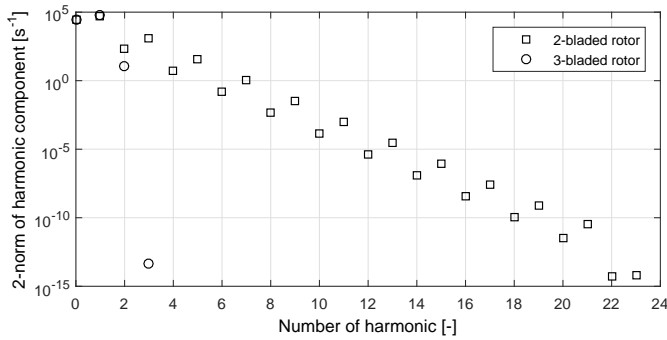

**Figure 3.** The 2-norms of the Fourier components of the system matrices for the 2- and 3-bladed simple models of the DTU 10MW RWT.

(9/rev) is 0.03 s$^{-1}$. For the 3-bladed rotor all $N = 3$ harmonic components are used although the third harmonic is effectively zero.



### 4.3 Campbell diagrams of 2- and 3-bladed turbines

The number of harmonics in the periodic eigenvectors used for Hill's truncated eigenvalue problem is set to $M = 2N$ for both rotors, thus $M = 14$ and $M = 6$ for the 2-and 3-bladed rotor, respectively. The eigenvalue problem is solved for 33 equidistant rotor speeds ranging from 2 rpm to 10 rpm (nominal speed of the DTU 10MW RWT is 9.6 rpm). The principal solutions at

the lowest speed of 2 rpm are selected among the sub-sets of solutions as the one that has the largest mean component on the ground-fixed DOFs (cf. Section 3.2). For the remaining rotor speeds, the principal solutions are selected as the ones that have the largest Modal Assurance Criterion relative to the principal solutions selected at the previous rotor speed.

    Figure 4 shows the Campbell diagrams of principal modal frequencies versus rotor speed for the 3-bladed (left) and 2-bladed (right) versions of the DTU 10MW RWT. The mode names are deduced from the analysis of the periodic mode shapes

presented in the next sections. For the 3-bladed rotor, the principal modal frequencies computed with Hill's method (circles) are in close agreement of the modal frequencies computed with the high-fidelity software HAWCStab2 (crosses). It shows that the simple model combined with the automatic identification of the principal solutions is able to predict the same modal frequencies in the ground-fixed frame as computed using the Coleman transformation method.

    Comparison of the two Campbell diagrams shows that the tower modes of the 2-bladed turbine have slightly higher fre-

quencies due to the lighter rotor. The $\pm 1$/rev frequencies splitting for the pairs of whirling rotor modes in the ground-fixed frame is clear for the 3-bladed turbine (noting that out-of-plane blade deflections are stiffened by the centrifugal forces). For the 2-bladed turbine, there are no whirling modes but the anti-symmetric modes are still either increasing (first edge and flap) or decreasing (second flap) with 1/rev, because the principal solutions are selected as the solutions with the largest mean components on ground-fixed DOFs. Campbell diagrams containing only the principal frequencies are not very informative for

2-bladed turbines, instead the periodic mode shapes of the selected principal solutions are now analyzed, first for the 3-bladed turbine. Note that first mode of both turbines is the trivial rigid body rotation of the drivetrain, which is omitted from this analysis.

### 4.4 Periodic mode shapes of 3-bladed turbines

Figures 5 – 15 show the dominating modal components of selected DOFs in the periodic mode shapes of Modes 2 – 12 of

the 3-bladed turbine as functions of rotor speed $\Omega$ and the frequency of the particular component $\omega_0 + m\Omega$, where $\omega_0$ is the principal modal frequency and $m$ is the harmonic order of the component (cf. Eq. (33)). A threshold for plotted amplitudes is set to 10 % of the overall maximum. The generator rotation and shaft torsion angles are multiplied by 10 for better scaling; the same scaling is used on the tower translations to show weak couplings for some modes. The plot layout is the same in all figures: A three-dimensional plot in the top and its projections onto the amplitude-rotor speed (left) and frequency-rotor speed

(right) planes in the lower plots. The principal frequencies $\omega_0$ are always denoted by black bullets in the frequency-rotor speed planes. The color coding of the modal components is also the same in all figures: Black markers denote amplitudes of ground-fixed DOFs. Red, green, and blue markers denote amplitudes of the rotor components in the first flapwise, first edgewise, and second flapwise deflection shapes, respectively. These rotor amplitudes are computed for each harmonic component of



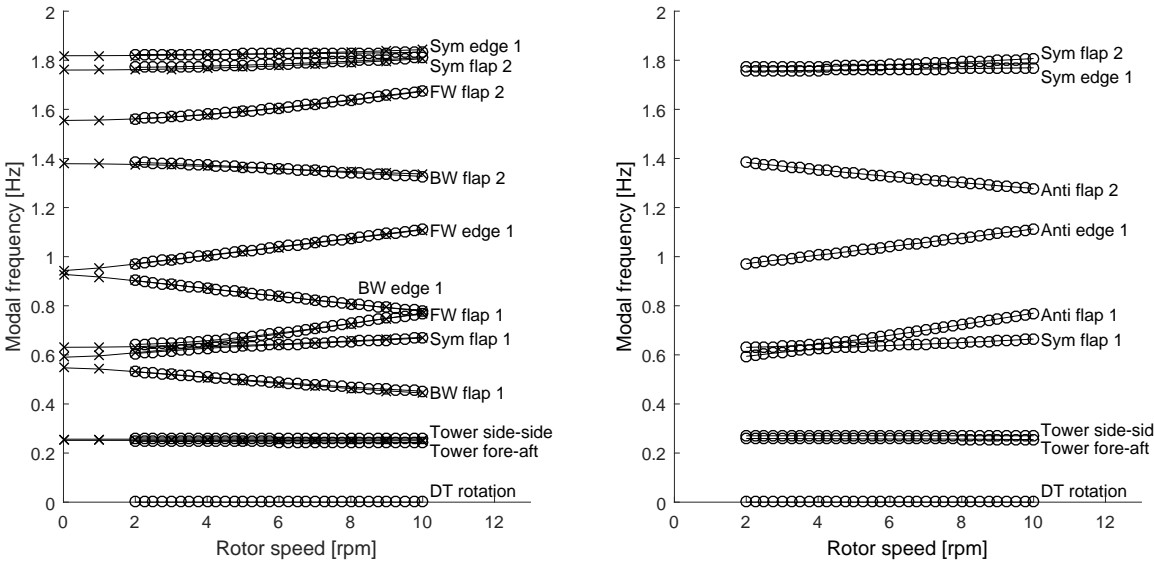

**Figure 4.** Campbell diagrams of the principal modal frequencies of the 3-bladed (left plot) and 2-bladed (right plot) version of the DTU 10MW RWT computed with Hill's method (circles) and with the software HAWCStab2 (crosses) for the 3-bladed turbine.

the periodic eigenvector using Eq. (38). Note that the frequency-rotor speed plane (lower right) are similar to the periodic Campbell diagrams in Bottasso & Cacciola (2015), except that each curve represent the magnitude of the harmonic component for a particular DOF and not the vector norm of all DOFs like the participation factors.

The naming of the modes shown in Figure 4 is deduced from the largest components of the periodic mode shapes observed
in Figures 5 – 15. Note that Modes 5 and 6, the first flapwise symmetric and forward whirling modes, interchange their mode shapes at low rotor speeds, which happens when the resolution of the rotor speed is so low that the sorting routine based on a Modal Assurance Criterion cannot distinguish these strongly coupled modes from each other.

Looking at all figures with, no harmonic component for the rotor whirling motion is more than ±1/rev from the principal frequency, and lowering threshold for the plotted amplitudes to 0.1 % of the overall maximum amplitude does not change
this observation. It agrees with Eq. (37) and previous studies of isotropic 3-bladed turbines using the Coleman transformation method (Hansen, 2003, 2007): an inverse Coleman transformation from the ground-fixed frame with a single modal frequency back to the rotating frame shows that symmetric components will have the same frequency as in the ground-fixed frame, but BW and FW components will have frequencies that are +1/rev and -1/rev, respectively, from a principal frequency defined in the ground-fixed frame. If the 3-bladed rotor would be anisotropic then secondary harmonics will arise which magnitudes will
depend on the magnitude of the anisotropy (Skjoldan & Hansen, 2009).

Looking at the individual mode shapes, there are couplings of the different DOFs in each mode of 3-bladed turbines. The tower fore-aft mode (Figure 5) contains symmetric, BW and FW components in the first flapwise blade mode. Similarly, the




tower side-side mode (Figure 6) contains BW and FW components in the first edgewise blade mode and generator rotation due to the nacelle roll. The first BW flapwise mode (Figure 7) is not pure and contains also symmetric and FW components. The strongly coupled first symmetric (Figure 8) and FW (Figure 9) flapwise modes are also not pure and contain other components as well. Note that a BW edgewise component (in green) appears in the first FW flapwise mode at the highest rotor speeds where

5    its BW flapwise component approaches the edgewise blade frequency of about 0.93 Hz. Similarly, the FW flapwise components appears in the first BW edgewise mode (Figure 10) at higher rotor speeds. The first FW edgewise mode (Figure 11) contains a small BW edgewise component and a larger FW flapwise component. Similar couplings can be seen for the remaining modes. The second flapwise whirling modes (Figure 12 and 13) have the largest blade response at a frequency of about 1.5 Hz (the +1/rev for BW and -1/rev for the FW components). The amplitude ratio of these flapwise DOFs are almost constant indicating

10    that the rotor modes are constructed as a combination of the two blade modes with the isolated modal frequencies 0.61 Hz and 1.74 Hz. Note that the first symmetric edgewise mode (Figure 15) couples with both the generator rotation and the shaft torsion. This rotor mode is neutral with respect to the rotor rotation and does therefore not contain any dominant whirling components. The second symmetric flapwise mode (Figure 14) contains whirling components that are just below the selected threshold for the plotted amplitudes and therefore not shown.

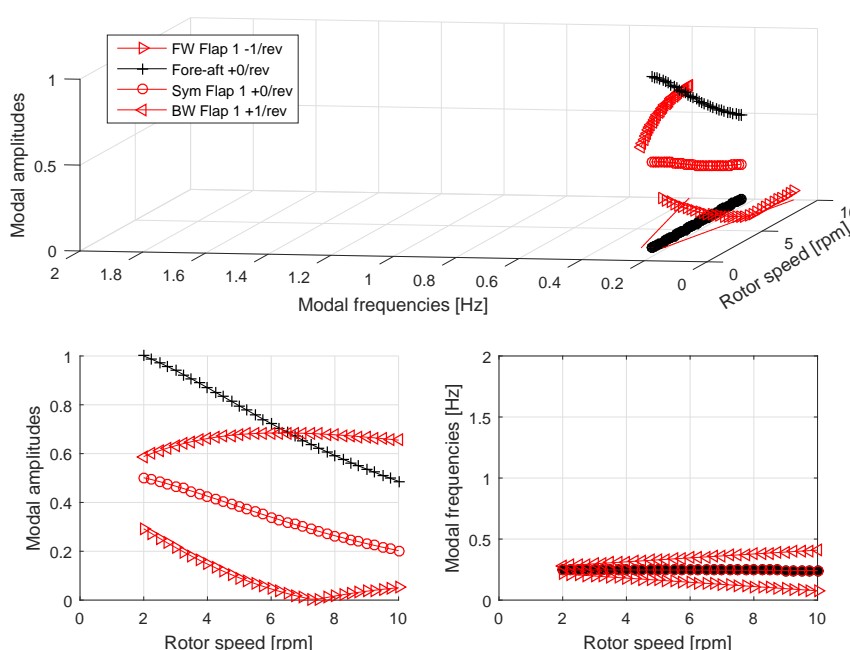

**Figure 5.** Dominating harmonic modal components for Mode 2 for the 3-bladed DTU 10MW RWT. Top plot: Modal amplitudes plotted versus rotor speed and frequency of the particular harmonic component. Lower left plot: Projection onto the plane of modal amplitudes and rotor speed. Lower right plot: Projection onto the plane of component frequencies and rotor speed (periodic Campbell diagram). Bullets show the principal modal frequencies in the frequencies and rotor speed planes. Only modal components with amplitudes larger than 10 % of the overall maximum amplitude are plotted.




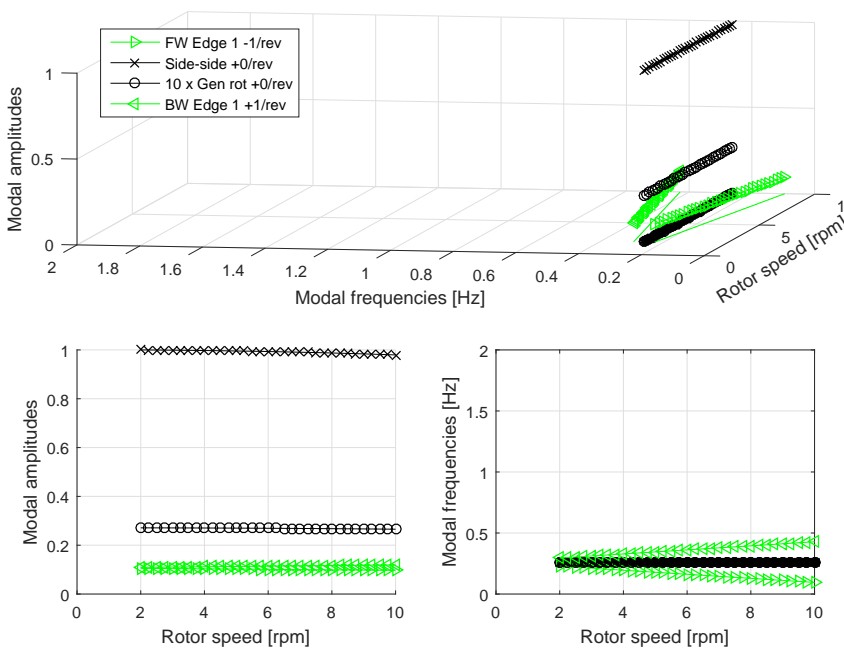

**Figure 6.** Dominating harmonic modal components for Mode 3 for the 3-bladed DTU 10MW RWT. Plot layout as in Figure 5.

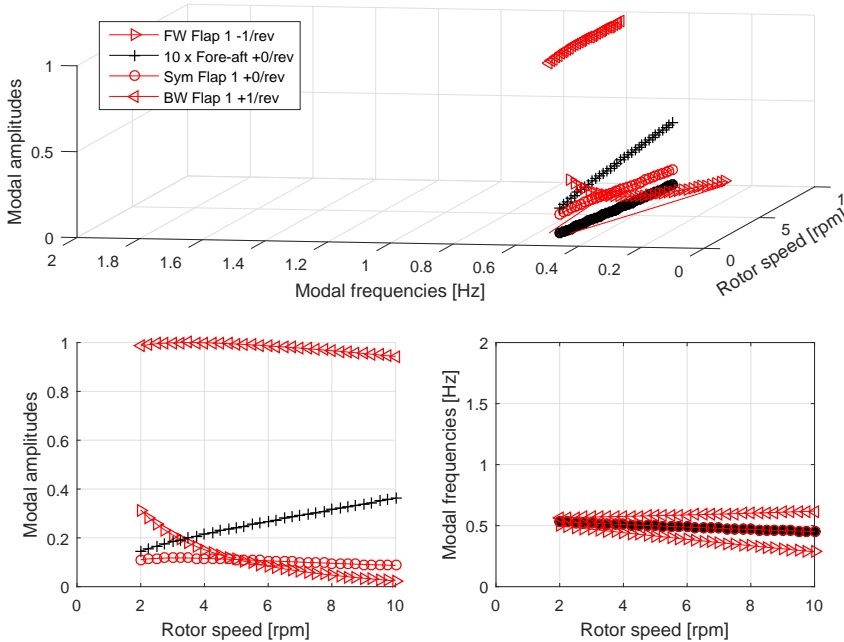

**Figure 7.** Dominating harmonic modal components for Mode 4 for the 3-bladed DTU 10MW RWT. Plot layout as in Figure 5.




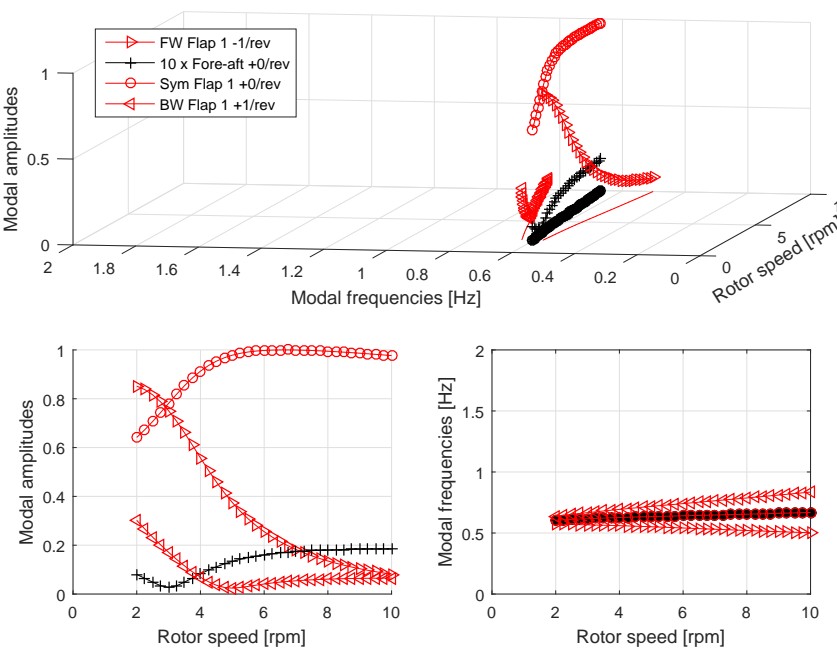

**Figure 8.** Dominating harmonic modal components for Mode 5 for the 3-bladed DTU 10MW RWT. Plot layout as in Figure 5.

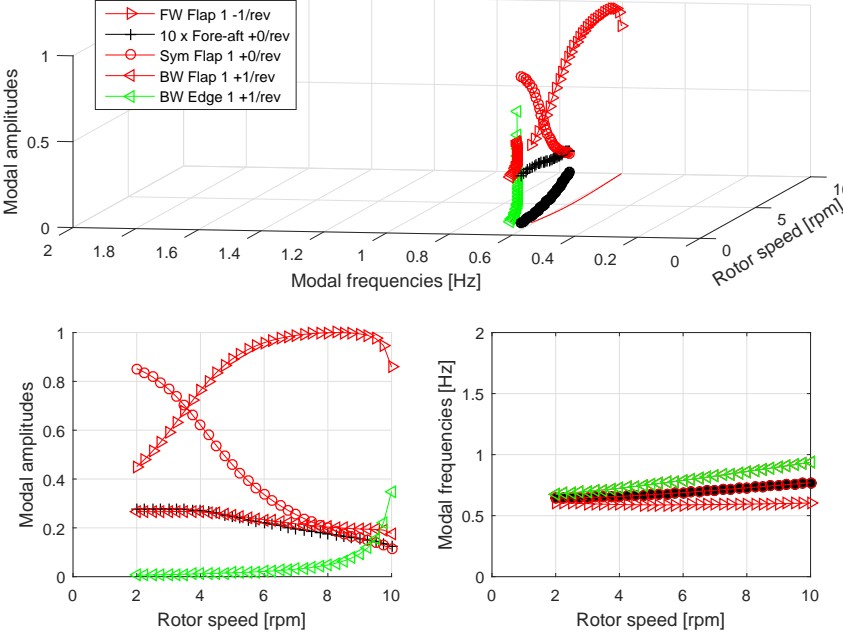

**Figure 9.** Dominating harmonic modal components for Mode 6 for the 3-bladed DTU 10MW RWT. Plot layout as in Figure 5.




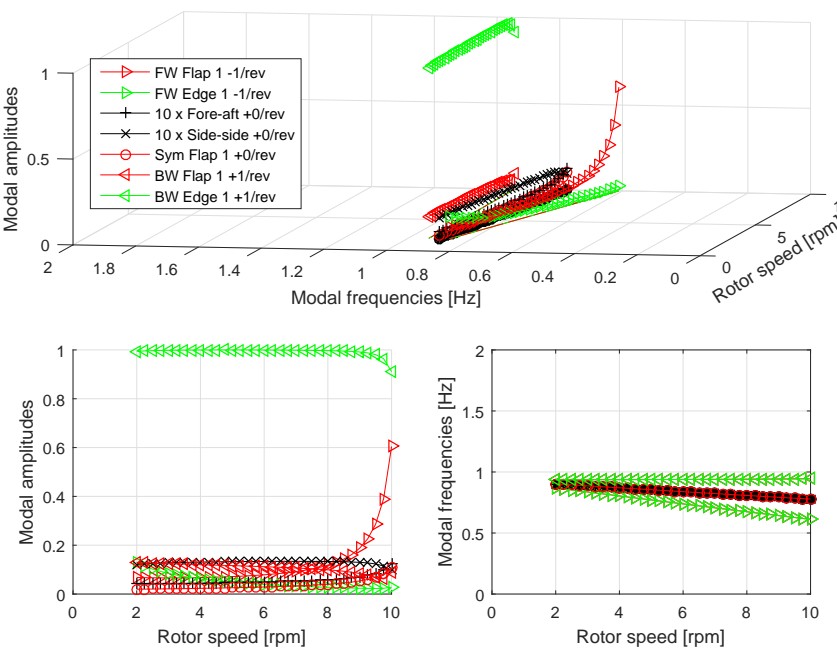

**Figure 10.** Dominating harmonic modal components for Mode 7 for the 3-bladed DTU 10MW RWT. Plot layout as in Figure 5.

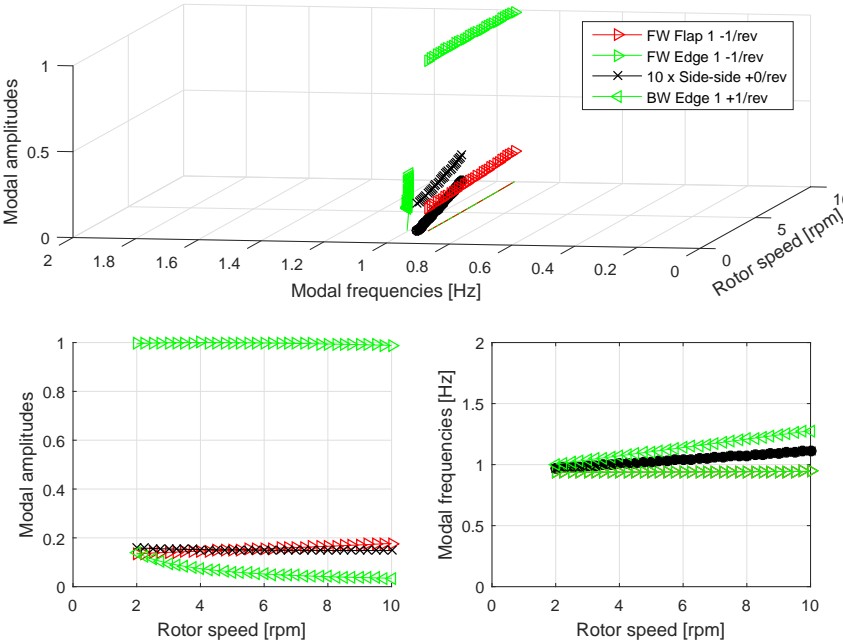

**Figure 11.** Dominating harmonic modal components for Mode 8 for the 3-bladed DTU 10MW RWT. Plot layout as in Figure 5.




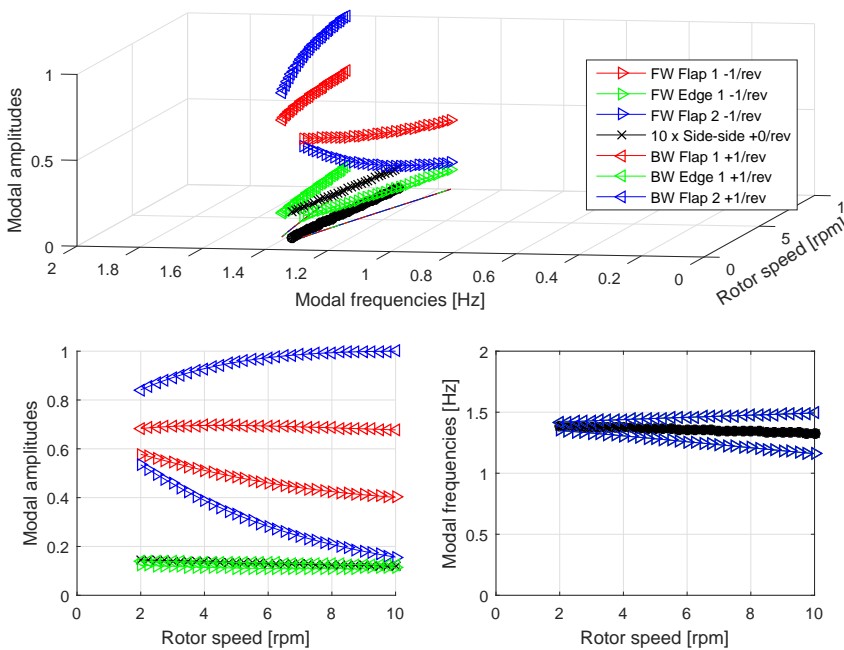

**Figure 12.** Dominating harmonic modal components for Mode 9 for the 3-bladed DTU 10MW RWT. Plot layout as in Figure 5.

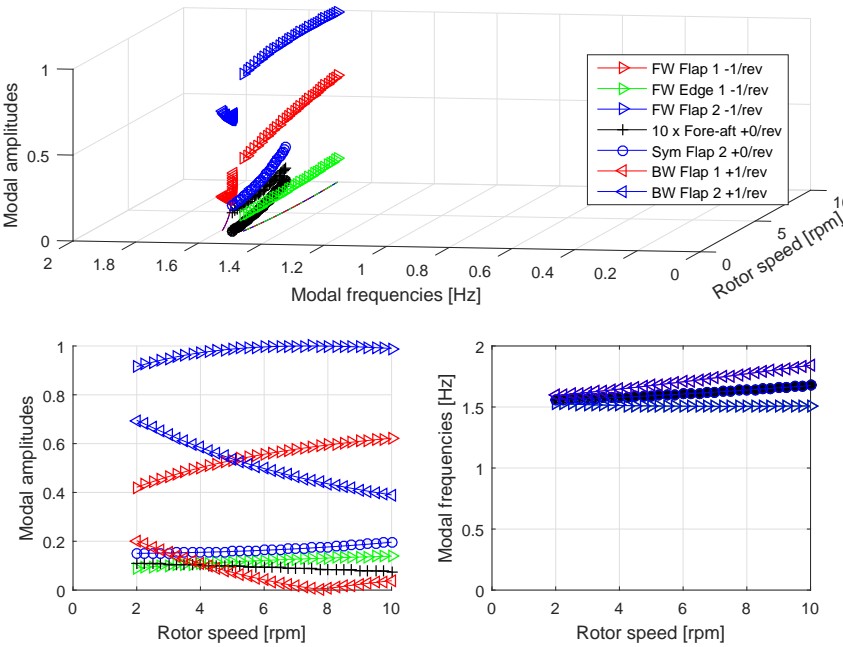

**Figure 13.** Dominating harmonic modal components for Mode 10 for the 3-bladed DTU 10MW RWT. Plot layout as in Figure 5.





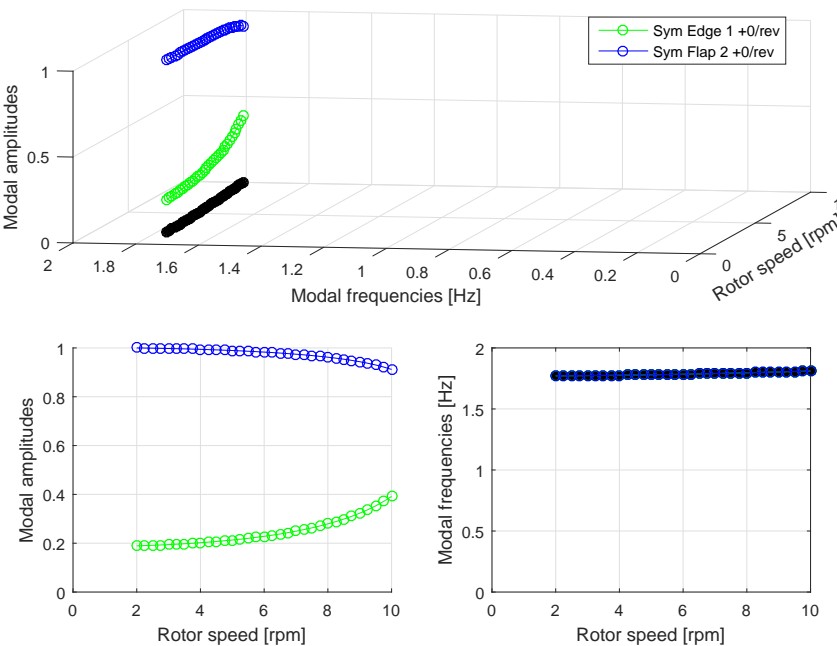

**Figure 14.** Dominating harmonic modal components for Mode 11 for the 3-bladed DTU 10MW RWT. Plot layout as in Figure 5.

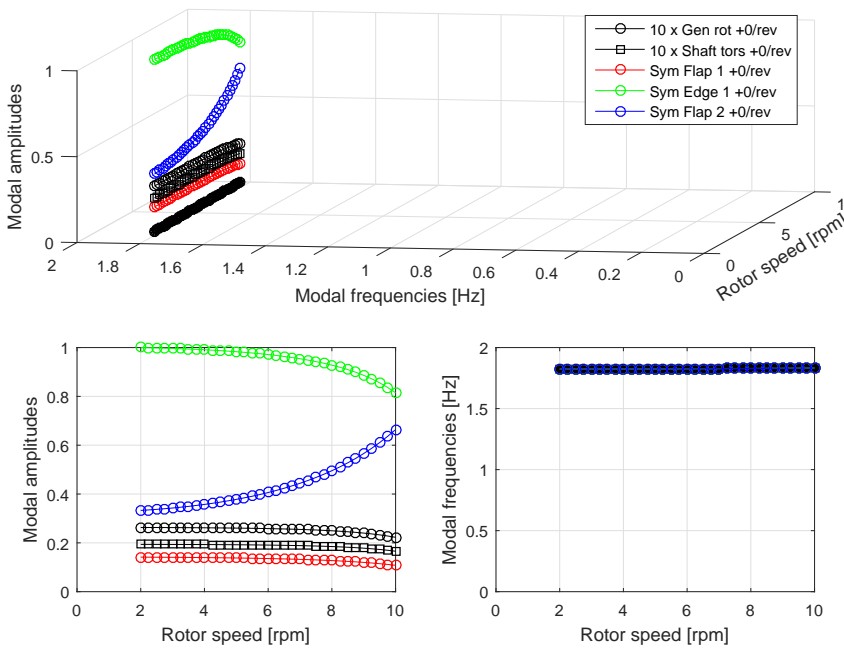

**Figure 15.** Dominating harmonic modal components for Mode 12 for the 3-bladed DTU 10MW RWT. Plot layout as in Figure 5.





### 4.5 Periodic mode shapes of 2-bladed turbines

Figures 16 – 23 show the dominating modal components of selected DOFs in the periodic mode shapes of Modes 2 – 9 of the 2-bladed turbine. The threshold for the plotted amplitudes is 10 % of the overall maximum amplitude. Generator rotation and shaft torsion angles are multiplied by 10 for better scaling, and the tower translations are also scaled to show weak couplings
in some modes. The plot layout and color coding are the same as above. The amplitudes of rotor motion are computed from the periodic eigenvector using Eq. (38).

Looking at all figures, there are many more dominating components and also higher harmonics compared to the 3-bladed turbine. A general observation is that symmetric rotor components always have frequencies that are even number of 1/rev away from the principal frequency, whereas frequencies for anti-symmetric components are shifted an odd number.

Looking at the individual mode shapes, the increased number of harmonics also increases the number of couplings between the different DOFs in each mode. The tower fore-aft mode in Figure 16 couples again with $\pm 1$/rev asymmetric rotor modes as for the 3-bladed turbine, but couplings to higher harmonic components of blade DOFs are much more dominant at rotor speeds where their frequency crosses the corresponding blade frequency. Such resonant couplings occur for the symmetric +4/rev and +6/rev components and anti-symmetric +3/rev and +5/rev components of the first flapwise DOF when their frequencies are
close to the first flapwise blade frequency 0.61 Hz, and for the anti-symmetric +5/rev component of the first edgewise DOF when its frequency crosses 0.93 Hz.

The tower side-side mode in Figure 17 is similar to the same mode of the 3-bladed turbine with the coupling to the generator rotation and the $\pm 1$/rev harmonics of asymmetric rotor responses.

Modes 4 and 5 in Figures 18 and 19 interchange mode shapes at low rotor speeds such that they are named the first symmetric
and anti-symmetric flapwise modes, respectively, at the higher rotor speeds. The anti-symmetric flapwise mode couples with tower fore-aft motion around 6.5 rpm where the frequency of the -4/rev component of this DOF is crossing the tower fore-aft frequency of about 0.28 Hz. A similar coupling is seen for the -6/rev component of the tower fore-aft DOF in the anti-symmetric edgewise mode in Figure 20 at 8.2 rpm. This mode also contain 0/rev and -2/rev components of the tower side-side DOF such that the $\pm 1$/rev splitting known from 3-bladed rotors appears in the ground-fixed frame.

For the second anti-symmetric flapwise mode in Figure 21, the anti-symmetric +1/rev components of the two flapwise blade DOFs are the two largest components with an almost constant frequency of around 1.46 Hz (a combination of the two flapwise blade modes as for the 3-bladed turbine). The blade motion also contains smaller anti-symmetric -1/rev and +3/rev components of the two flapwise blade DOFs, which frequencies are decreasing and increasing with 2/rev in the rotating blade frame. Resonant couplings with symmetric +6, +8, and +10/rev second flapwise DOF and +6/rev first edgewise DOF are
observed when their frequencies are crossing the principal frequencies of the first symmetric edgewise and second symmetric flapwise turbine modes around 1.75 Hz. These two symmetric modes in Figures 22 and 23 are similar to the corresponding modes of the 3-bladed turbine (cf. Figures 14 and 15), except that the second symmetric flapwise mode of the 2-bladed turbine contains resonant couplings of the anti-symmetric -3/rev and -5/rev components of the two flapwise blade DOFs when their frequencies are crossing the frequency of the second anti-symmetric flapwise mode (cf. Figure 21) around 1.46 Hz.




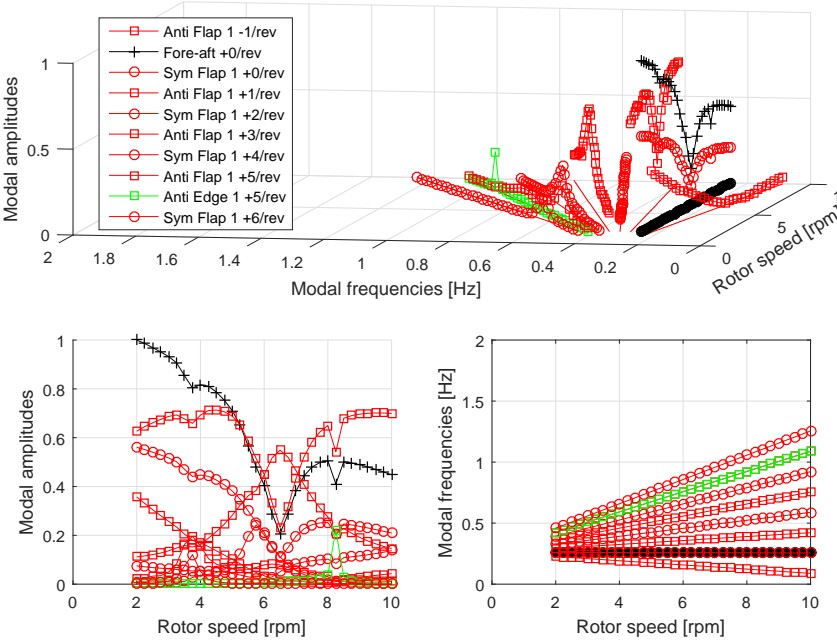

**Figure 16.** Dominating harmonic modal components for Mode 2 for the 2-bladed version of the DTU 10MW RWT. Plot layout as in Figure 5.

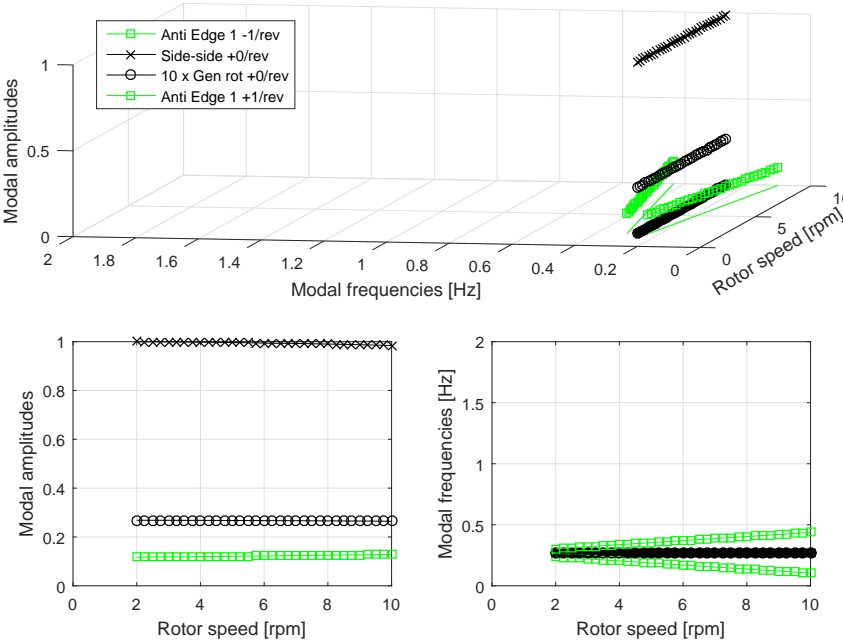

**Figure 17.** Dominating harmonic modal components for Mode 3 for the 2-bladed version of the DTU 10MW RWT. Plot layout as in Figure 5.





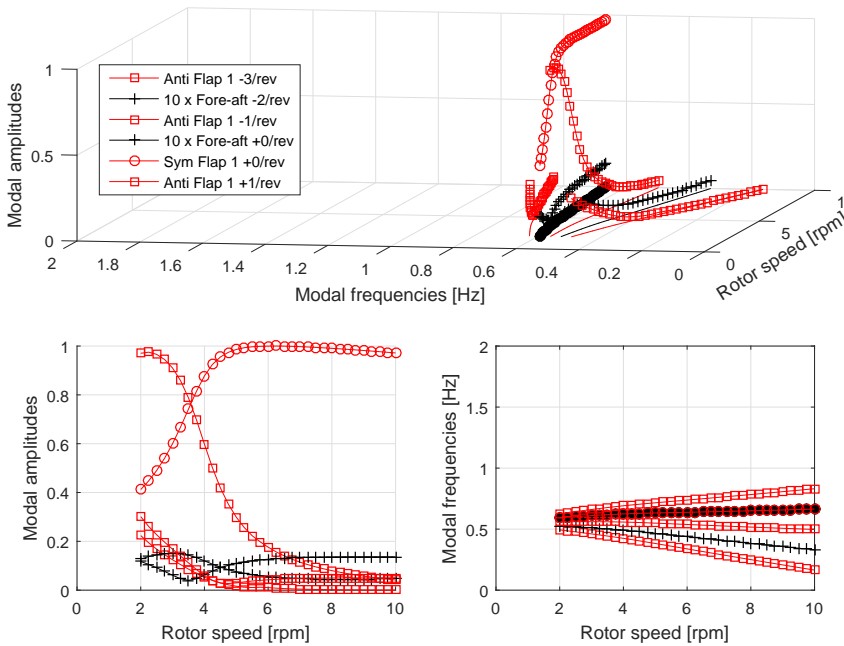

**Figure 18.** Dominating harmonic modal components for Mode 4 for the 2-bladed version of the DTU 10MW RWT. Plot layout as in Figure 5.

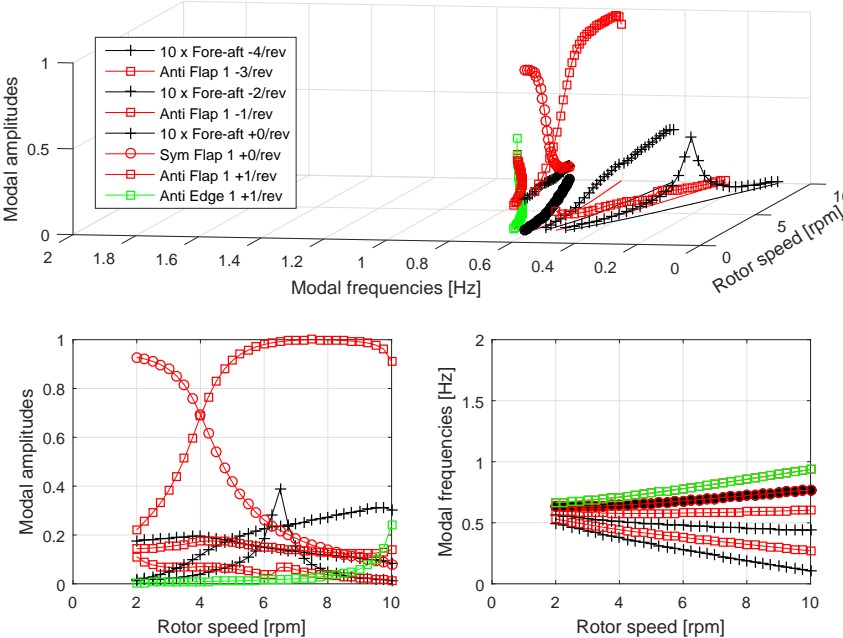

**Figure 19.** Dominating harmonic modal components for Mode 5 for the 2-bladed version of the DTU 10MW RWT. Plot layout as in Figure 5.



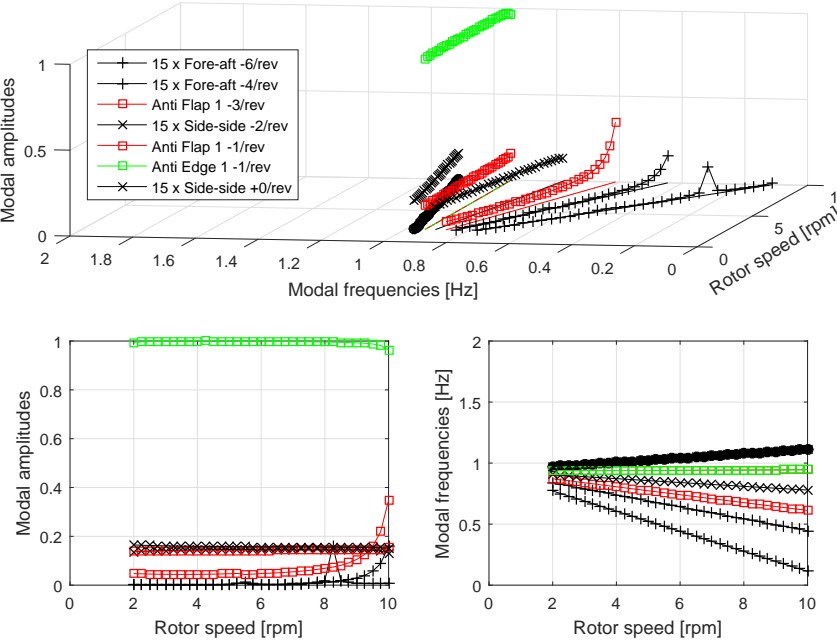

**Figure 20.** Dominating harmonic modal components for Mode 6 for the 2-bladed version of the DTU 10MW RWT. Plot layout as in Figure 5.

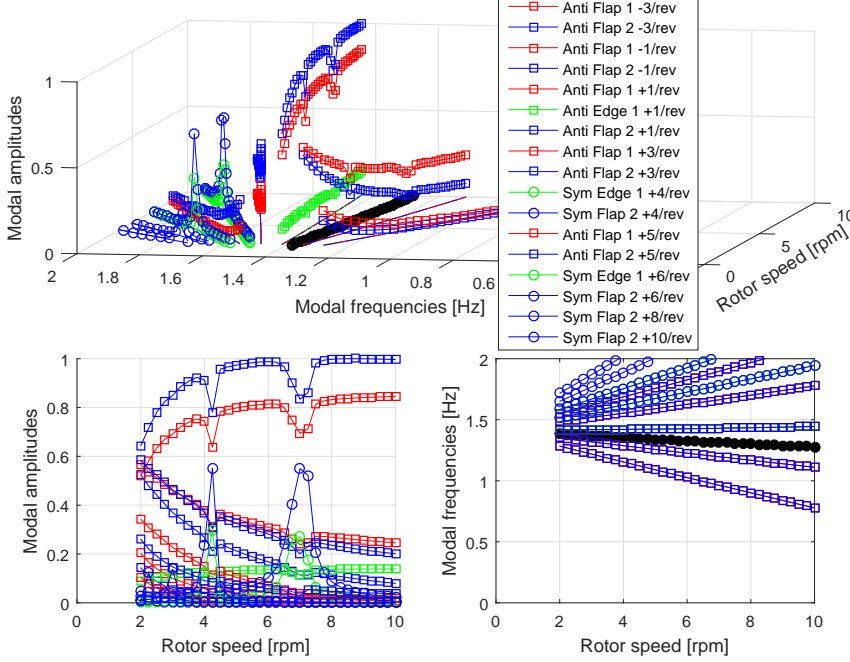

**Figure 21.** Dominating harmonic modal components for Mode 7 for the 2-bladed version of the DTU 10MW RWT. Plot layout as in Figure 5.




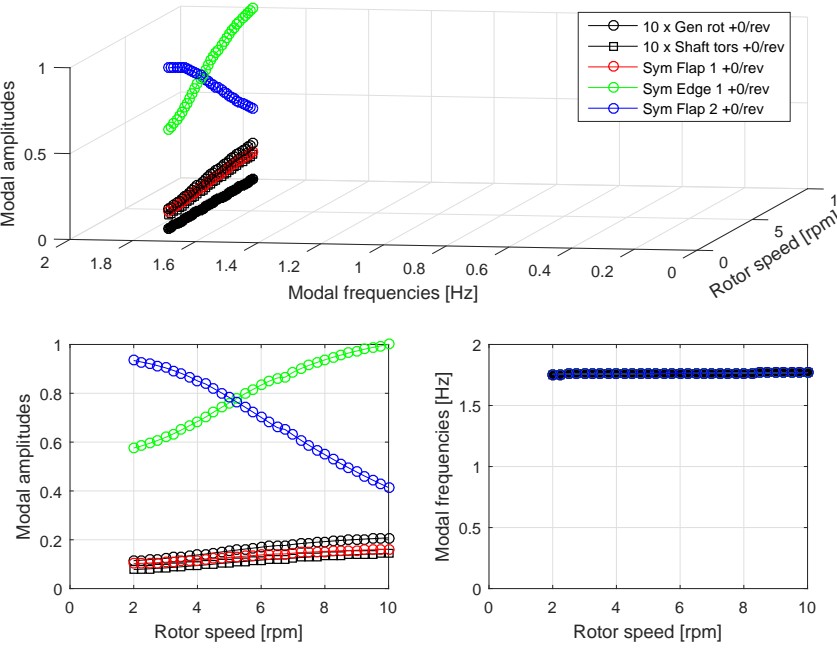

**Figure 22.** Dominating harmonic modal components for Mode 8 for the 2-bladed version of the DTU 10MW RWT. Plot layout as in Figure 5.

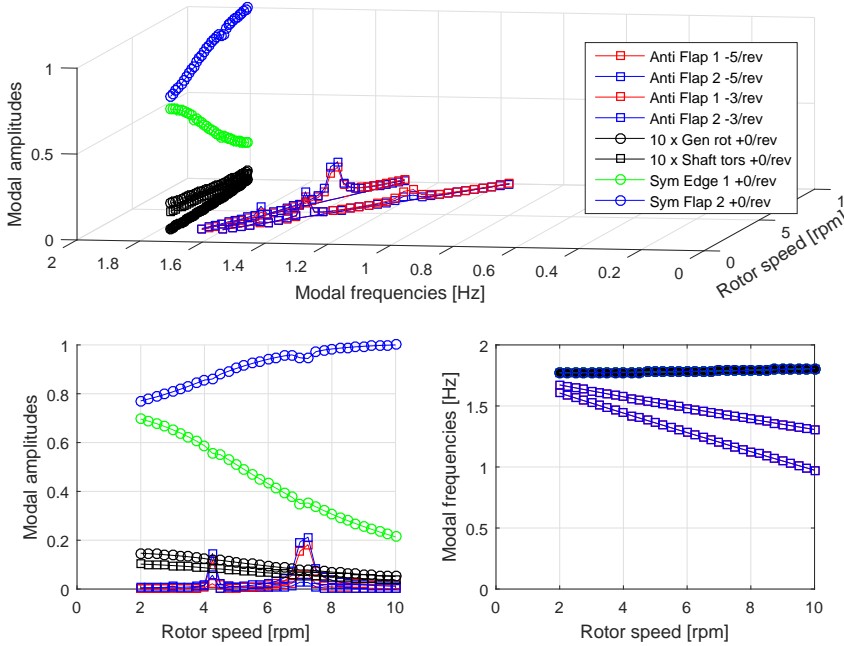

**Figure 23.** Dominating harmonic modal components for Mode 9 for the 2-bladed version of the DTU 10MW RWT. Plot layout as in Figure 5.





### 4.6 Comparison of modal dynamics of 2- and 3-bladed turbines

The modal dynamics of the turbines with 2- and 3-bladed rotors show several similarities but also significant differences:

- The rigid body drivetrain rotation mode (trivial and therefore not shown) is identical for the two turbines.

- Tower bending modes are similar in frequencies and shapes, except that the fore-aft mode for the 2-bladed turbine may contain large components of the first flapwise blade mode when the rotor speed is such that these frequencies of the higher harmonic components are crossing the modal frequency of this blade mode.

- The first symmetric edgewise rotor mode are very similar in frequency and shape because its reaction forces does not coupled to other DOFs through large periodic terms in the system matrix.

- The symmetric flapwise rotor modes are similar in frequencies and shapes, except that the first symmetric flapwise mode of the 2-bladed turbine in Figure 18 has a small -2/rev component of the tower fore-aft DOF, and the second symmetric flapwise mode in Figure 23 has resonant couplings to anti-symmetric flapwise mode.

- Asymmetric rotor modes: the anti-symmetric modes for the 2-bladed turbine and the whirling mode pairs of the 3-bladed turbine, may seem similar when observed from the ground-fixed frame such as a top tower acceleration signal, where the well-known ±1/rev splitting of the frequency peaks is seen. For example, the tower side-side responses at ±1/rev around the blade edgewise frequency is observed for both the anti-symmetric edgewise mode in Figure 20 and for the edgewise whirling mode pair in Figures 10 and 11. This similarity has probably caused the misinterpretation of the simulated and measured 2-bladed responses in Kim et al. (2015); Larsen & Kim (2015), but rotor modes of a 2-bladed turbine will often have more frequency peaks in both the ground-fixed frame and the rotating blade frame.

The additional modal couplings that exist for 2-bladed turbines have the effects that there are additional ways that the modes can be excited either by resonances or interactions with external forces, and that it becomes difficult to interpret frequency spectra from simulations and measurements. To illustrate these effects, the harmonic components of the tower side-side DOF in all modes are plotted in three-dimensional periodic Campbell diagrams for both turbines in Figure 24. These plots show qualitatively the response in a tower side-side signal when all modes are excited equally. There are clear similarities between the dominant side-side responses for the two turbines. But where these responses in the ground-fixed frame for the 3-bladed turbine only occur on the principal modal frequencies plotted in its conventional Campbell diagram (cf. Figure 4), the modes of 2-bladed turbine lead to responses in the ground-fixed frame at many other frequencies. The conventional Campbell diagram containing only the principal modal frequencies of a 2-bladed turbine has therefore little value, and one should be careful when interpreting frequency response plots for 2-bladed turbines.

## 5 Conclusions

The modal dynamics of structures with bladed isotropic rotors (identical and equidistantly spaced blades) have been analyzed by the periodic mode shapes obtained using Hill's method on the linear periodic first order system equation. Analytical deriva-





tion of linear second order equations of motion in a generic form has shown that only 1/rev harmonics occur in the periodic terms when a rotor has more than two blades, whereas a 2-bladed rotor also has 2/rev terms. Analytical inversion of the periodic mass matrix has shown that its highest harmonic term 2/rev for an isotropic rotor with more than two blades. The inverse mass matrix for a 2-bladed rotor have been shown to have an infinite Fourier series of component matrices which norm decreases

with the harmonic order. The periodic system matrix of isotropic rotors with more than two blades can therefore be represented by an exact Fourier series with 3/rev being the highest order, whereas it for 2-bladed rotors must be approximated by a truncated Fourier series of predictable accuracy.

Using the analytic Fourier series of the system matrix to set up Hill's truncated eigenvalue problem, its principal solutions have been automatically identified by a novel method applicable to larger systems. The symmetric and anti-symmetric modal

components of rotors with two blades and the additional whirling components of rotors with more blades have been extracted directly from the periodic eigenvector corresponding to each principal eigen-solution.

As relevant examples, the generic methods have been used to model and analyze the modal dynamics of both 2- and 3-bladed versions of a 10MW turbine. The motion of each blade has been described by its three first mode shapes, and the nacelle motion and drivetrain rotations have been described by seven DOFs. Similarities and significant differences between

the modal dynamics of the turbines with 2- and 3-bladed rotors have been found and summarized in Section 4.6. A difference is that the larger number and magnitudes of the harmonic components in the system matrix of 2-bladed turbines lead to resonant couplings in the mode shapes that have not been observed for the 3-bladed turbine. These couplings between DOFs arise when the rotor speed is such that the frequency of a higher harmonic component of one turbine mode is in the vicinity of the frequency of another blade or turbine mode. Another difference is that a single mode of a 2-bladed turbine can lead to responses

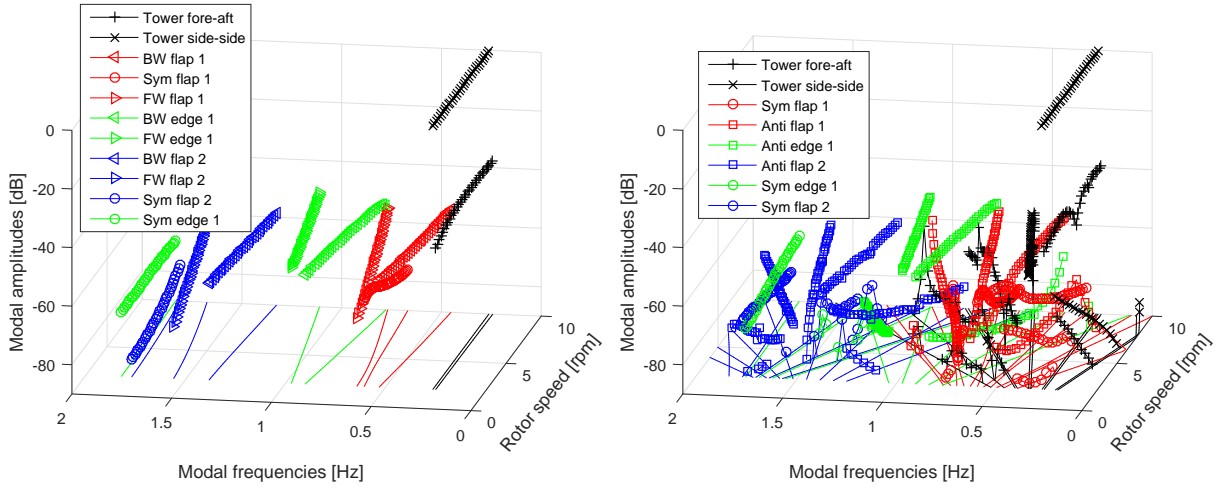

**Figure 24.** Periodic Campbell diagrams of all harmonic components of the tower side-side DOF in all investigated modes for the 3-bladed (left plot) and 2-bladed (right plot) version of the DTU 10MW RWT.



at several frequencies in both the ground-fixed and rotating blade frames of reference, which complicates the interpretation of simulated or measured turbine responses without the theories presented in this paper.

## Appendix A: Elements of system matrices

This appendix contains a list of the elements of the block matrices in (13) and (14) related to the inertia forces and derived

by insertion of (10), (8) and (9) into the coefficients (7), and summation over all volumes of the structure, the ground-fixed substructure $\mathcal{V}_g$ and each blade $\mathcal{V}_b$, where several properties of the rotation matrix $\mathbf{R}$ and its components have been utilized (see e.g. Krenk (2009)). The subscripts $g$ and $b$ are omitted from the DOFs $u_i$ and $u_j$ for brevity.

The elements of the block matrices for the blade DOFs are

$$m_{b,ij} = \int_{\mathcal{V}_b} \rho \frac{\partial \mathbf{r}_b^T}{\partial u_i} \frac{\partial \mathbf{r}_b}{\partial u_j} d\mathcal{V}_b$$

$$g_{b,ij} = 4\Omega \int_{\mathcal{V}_b} \rho \frac{\partial \mathbf{r}_b^T}{\partial u_i} \mathbf{B}_1 \frac{\partial \mathbf{r}_b}{\partial u_j} d\mathcal{V}_b \tag{A1}$$

$$s_{b,ij} = -\Omega^2 \int_{\mathcal{V}_b} \rho \left( \frac{\partial^2 \mathbf{r}_b^T}{\partial u_i \partial u_j} \mathbf{A}_1 \mathbf{r}_b + \frac{\partial \mathbf{r}_b^T}{\partial u_i} \mathbf{A}_1 \frac{\partial \mathbf{r}_b}{\partial u_j} \right) d\mathcal{V}_b$$

where $\mathbf{A}_1 = \frac{1}{2} \left( \mathbf{R}_1 - \bar{\mathbf{R}}_1 \right)$ and $\mathbf{B}_1 = \frac{1}{2} \left( \mathbf{R}_1 + \bar{\mathbf{R}}_1 \right)$ are the symmetric real part and the anti-symmetric imaginary part of the matrix $\mathbf{R}_1$. Note that $\mathbf{R}_0^T = \mathbf{R}_0$ and $\bar{\mathbf{R}}_1 = \mathbf{R}_1^T$. The elements of the block matrices for the DOFs of the ground-fixed substructure are

$$m_{g,0,ij} = \int_{\mathcal{V}_g} \rho \frac{\partial \mathbf{r}_g^T}{\partial u_i} \frac{\partial \mathbf{r}_g}{\partial u_j} d\mathcal{V}_g$$

$$+ B \int_{\mathcal{V}_b} \rho \left( \frac{\partial \mathbf{r}_c^T}{\partial u_i} \frac{\partial \mathbf{r}_c}{\partial u_j} + \mathbf{r}_b^T \mathbf{R}_0 \left( \frac{\partial \mathbf{T}_c^T}{\partial u_i} \frac{\partial \mathbf{r}_c}{\partial u_j} + \frac{\partial \mathbf{T}_c^T}{\partial u_j} \frac{\partial \mathbf{r}_c}{\partial u_i} \right) \right.$$

$$\left. + \mathbf{r}_b^T \left( \mathbf{R}_0 \frac{\partial \mathbf{T}_c^T}{\partial u_i} \frac{\partial \mathbf{T}_c}{\partial u_j} \mathbf{R}_0 + \mathbf{R}_1 \left( \frac{\partial \mathbf{T}_c^T}{\partial u_i} \frac{\partial \mathbf{T}_c}{\partial u_j} + \frac{\partial \mathbf{T}_c^T}{\partial u_j} \frac{\partial \mathbf{T}_c}{\partial u_i} \right) \mathbf{R}_1 \right) \mathbf{r}_b \right) d\mathcal{V}_b$$

$$m_{g,2,ij} = \begin{cases} 2 \int_{\mathcal{V}_b} \rho \mathbf{r}_b^T \mathbf{R}_1^T \frac{\partial \mathbf{T}_c^T}{\partial u_i} \frac{\partial \mathbf{T}_c}{\partial u_j} \mathbf{R}_1 \mathbf{r}_b d\mathcal{V}_b & \text{for } B = 2 \\ 0 & \text{otherwise} \end{cases} \tag{A2}$$

$$g_{g,0,ij} = \imath 2\Omega B \int_{\mathcal{V}_b} \rho \mathbf{r}_b^T \mathbf{R}_1 \left( \frac{\partial \mathbf{T}_c^T}{\partial u_i} \frac{\partial \mathbf{T}_c}{\partial u_j} - \frac{\partial \mathbf{T}_c^T}{\partial u_j} \frac{\partial \mathbf{T}_c}{\partial u_i} \right) \mathbf{R}_1 \mathbf{r}_b d\mathcal{V}_b$$

$$g_{g,2,ij} = \begin{cases} \imath 4\Omega \int_{\mathcal{V}_b} \rho \mathbf{r}_b^T \mathbf{R}_1^T \frac{\partial \mathbf{T}_c^T}{\partial u_i} \frac{\partial \mathbf{T}_c}{\partial u_j} \mathbf{R}_1 \mathbf{r}_b d\mathcal{V}_b & \text{for } B = 2 \\ 0 & \text{otherwise} \end{cases}$$





The elements of the block matrices that couples the blade and ground-fixed DOFs are

$$
m_{gb,0,ij} = \int_{\mathcal{V}_b} \rho \left( \frac{\partial \mathbf{r}_c^T}{\partial u_i} \mathbf{T}_c \mathbf{R}_0 \frac{\partial \mathbf{r}_b}{\partial u_j} \right.
$$

$$
\left. + \mathbf{r}_b^T \left( \mathbf{R}_0 \frac{\partial \mathbf{T}_c^T}{\partial u_i} \mathbf{T}_c \mathbf{R}_0 + \mathbf{R}_1^T \frac{\partial \mathbf{T}_c^T}{\partial u_i} \mathbf{T}_c \mathbf{R}_1^T + \mathbf{R}_1 \frac{\partial \mathbf{T}_c^T}{\partial u_i} \mathbf{T}_c \mathbf{R}_1 \right) \frac{\partial \mathbf{r}_b}{\partial u_j} \right) d\mathcal{V}_b \tag{A3}
$$

$$
m_{gb,1,ij} = \int_{\mathcal{V}_b} \rho \left( \left( \frac{\partial \mathbf{r}_c^T}{\partial u_i} \mathbf{T}_c \mathbf{R}_1 + \mathbf{r}_b^T \left( \mathbf{R}_0 \frac{\partial \mathbf{T}_c^T}{\partial u_i} \mathbf{T}_c \mathbf{R}_1 + \mathbf{R}_1^T \frac{\partial \mathbf{T}_c^T}{\partial u_i} \mathbf{T}_c \mathbf{R}_0 \right) \right) \frac{\partial \mathbf{r}_b}{\partial u_j} \right) d\mathcal{V}_b
$$

$$
g_{gb,1,ij} = \imath 2\Omega \int_{\mathcal{V}_b} \rho \left( \frac{\partial \mathbf{r}_c^T}{\partial u_i} \mathbf{T}_c \mathbf{R}_1 + \mathbf{r}_b^T \mathbf{R}_0 \frac{\partial \mathbf{T}_c^T}{\partial u_i} \mathbf{T}_c \mathbf{R}_1 \right) \frac{\partial \mathbf{r}_b}{\partial u_j} d\mathcal{V}_b \tag{A4}
$$

$$
g_{bg,0,ij} = \imath 2\Omega \int_{\mathcal{V}_b} \rho \left( \mathbf{r}_b^T \mathbf{R}_1^T \frac{\partial \mathbf{T}_c^T}{\partial u_j} \mathbf{T}_c \mathbf{R}_1 \frac{\partial \mathbf{r}_b}{\partial u_i} - \mathbf{r}_b^T \mathbf{R}_1 \frac{\partial \mathbf{T}_c^T}{\partial u_j} \mathbf{T}_c \mathbf{R}_1 \frac{\partial \mathbf{r}_b}{\partial u_i} \right) d\mathcal{V}_b
$$

$$
g_{bg,1,ij} = \imath 2\Omega \int_{\mathcal{V}_b} \rho \frac{\partial \mathbf{r}_b^T}{\partial u_i} \mathbf{R}_0 \mathbf{T}_c^T \frac{\partial \mathbf{T}_c}{\partial u_j} \mathbf{R}_1 \mathbf{r}_b d\mathcal{V}_b
$$

$$
s_{gb,1,ij} = \Omega^2 \int_{\mathcal{V}_b} \rho \left( \mathbf{r}_b^T \mathbf{R}_1^T \frac{\partial \mathbf{T}_c^T}{\partial u_i} \mathbf{T}_c \mathbf{R}_0 - \frac{\partial \mathbf{r}_c^T}{\partial u_i} \mathbf{T}_c \mathbf{R}_1 - \mathbf{r}_b^T \mathbf{R}_0 \frac{\partial \mathbf{T}_c^T}{\partial u_i} \mathbf{T}_c \mathbf{R}_1 \right) \frac{\partial \mathbf{r}_b}{\partial u_j} d\mathcal{V}_b
$$

**Appendix B: Components of inverted mass matrix**

The mean and harmonic components of the block matrices (27) of the inverse mass matrix (15) can be computed as

$$
\mathbf{E}_0 = \mathbf{M}_r^{-1} + \mathbf{M}_r^{-1} \Big( \mathbf{M}_{gr,1}^H \mathbf{H}_2 \bar{\mathbf{M}}_{gr,1}
$$

$$
+ \mathbf{M}_{gr,0}^T \mathbf{H}_0 \mathbf{M}_{gr,0} + \mathbf{M}_{gr,1}^H \mathbf{H}_0 \mathbf{M}_{gr,1} + \mathbf{M}_{gr,1}^T \mathbf{H}_0 \bar{\mathbf{M}}_{gr,1}
$$

$$
+ \mathbf{M}_{gr,1}^T \bar{\mathbf{H}}_2 \mathbf{M}_{gr,1} \Big) \mathbf{M}_r^{-1}
$$

$$
\mathbf{E}_{2m+1} = \mathbf{M}_r^{-1} \Big( \mathbf{M}_{gr,0}^T \mathbf{H}_{2m+2} \bar{\mathbf{M}}_{gr,1} + \mathbf{M}_{gr,1}^H \mathbf{H}_{2m+2} \mathbf{M}_{gr,0}
$$

$$
+ \mathbf{M}_{gr,0}^T \mathbf{H}_{2m} \mathbf{M}_{gr,1} + \mathbf{M}_{gr,1}^T \mathbf{H}_{2m} \mathbf{M}_{gr,0} \Big) \mathbf{M}_r^{-1} \tag{B1}
$$

$$
\mathbf{E}_{2m+2} = \mathbf{M}_r^{-1} \Big( \mathbf{M}_{gr,1}^H \mathbf{H}_{2m+4} \bar{\mathbf{M}}_{gr,1}
$$

$$
+ \mathbf{M}_{gr,0}^T \mathbf{H}_{2m+2} \mathbf{M}_{gr,0} + \mathbf{M}_{gr,1}^H \mathbf{H}_{2m+2} \mathbf{M}_{gr,1} + \mathbf{M}_{gr,1}^T \mathbf{H}_{2m+2} \bar{\mathbf{M}}_{gr,1}
$$

$$
+ \mathbf{M}_{gr,1}^T \mathbf{H}_{2m} \mathbf{M}_{gr,1} \Big) \mathbf{M}_r^{-1}
$$

and

$$
\mathbf{F}_{2m} = - \mathbf{H}_{2m} \mathbf{M}_{gr,0} \mathbf{M}_r^{-1}
$$

$$
\mathbf{F}_{2m+1} = - \mathbf{H}_{2m+2} \bar{\mathbf{M}}_{gr,1} \mathbf{M}_r^{-1} - \mathbf{H}_{2m} \mathbf{M}_{gr,1} \mathbf{M}_r^{-1} \tag{B2}
$$





where $m = 0, 1, \ldots, N_h$, $(\cdot)^H$ denotes the conjugate transpose matrix operator, the mean and harmonic components of the block matrix $\mathbf{H}$ (19) can be found in (21) and (22), and the conditions $\mathbf{H}_{2m} = \mathbf{0}$ for $m > N_H$ must be used. Note that $\mathbf{E}_{-n} = \bar{\mathbf{E}}_n$ and $\mathbf{F}_{-n} = \bar{\mathbf{F}}_n$.

**Appendix C: Components of system matrix**

The components of the system matrix (28) can be written as

$$\mathbf{A}_n = \begin{bmatrix} \mathbf{0} & & \mathbf{A}_{12} & \\ \mathbf{A}_{21,11,n} & \mathbf{A}_{21,12,n} & \mathbf{A}_{22,11,n} & \mathbf{A}_{22,12,n} \\ \mathbf{A}_{21,21,n} & \mathbf{A}_{21,22,n} & \mathbf{A}_{22,21,n} & \mathbf{A}_{22,22,n} \end{bmatrix} \tag{C1}$$

where $\mathbf{A}_{12} = \mathbf{I}$ for $n = 0$ and $\mathbf{A}_{12} = \mathbf{0}$ for $n > 0$, and the block matrices can be computed as

$$\mathbf{A}_{21,11,n} = -\mathbf{E}_n \left( \mathbf{K}_r + \mathbf{S}_r \right) - \mathbf{F}_{n-1}^T \mathbf{S}_{gr,1} - \mathbf{F}_{n+1}^T \bar{\mathbf{S}}_{gr,1}$$

$$\mathbf{A}_{21,21,n} = -\mathbf{F}_n \left( \mathbf{K}_r + \mathbf{S}_r \right) - \mathbf{H}_{n-1}^T \mathbf{S}_{gr,1} - \mathbf{H}_{n+1}^T \bar{\mathbf{S}}_{gr,1}$$

$$\mathbf{A}_{21,12,n} = -\mathbf{F}_n^T \mathbf{K}_g$$

$$\mathbf{A}_{21,22,n} = -\mathbf{H}_n \mathbf{K}_g \tag{C2}$$

$$\mathbf{A}_{22,11,n} = -\mathbf{E}_n \mathbf{C}_r - \mathbf{F}_n^T \mathbf{G}_{gr,0} - \mathbf{F}_{n-1}^T \mathbf{G}_{gr,1} - \mathbf{F}_{n+1}^T \bar{\mathbf{G}}_{gr,1}$$

$$\mathbf{A}_{22,21,n} = -\mathbf{F}_n \mathbf{C}_r - \mathbf{H}_n \mathbf{G}_{gr,0} - \mathbf{H}_{n-1} \mathbf{G}_{gr,1} - \mathbf{H}_{n+1} \bar{\mathbf{G}}_{gr,1}$$

$$\mathbf{A}_{22,12,n} = -\mathbf{F}_n^T \left( \mathbf{G}_{g,0} + \mathbf{C}_g \right) - \mathbf{E}_{n-1} \mathbf{G}_{rg,1} - \mathbf{E}_{n+1} \bar{\mathbf{G}}_{rg,1}$$

$$\qquad - \mathbf{F}_{n-2}^T \mathbf{G}_{g,2} - \mathbf{F}_{n+2}^T \bar{\mathbf{G}}_{g,2}$$

$$\mathbf{A}_{22,22,n} = -\mathbf{H}_n \left( \mathbf{G}_{g,0} + \mathbf{C}_g \right) - \mathbf{F}_{n-1} \mathbf{G}_{rg,1} - \mathbf{F}_{n+1} \bar{\mathbf{G}}_{rg,1}$$

$$\qquad - \mathbf{H}_{n-2} \mathbf{G}_{g,2} - \mathbf{H}_{n+2} \bar{\mathbf{G}}_{g,2}$$

for $n = 0, 1, \ldots, 2N_h + 3$ using the following properties of the matrices $\mathbf{E}_n$, $\mathbf{F}_n$, and $\mathbf{H}_n$:

$$\mathbf{E}_{-n} = \bar{\mathbf{E}}_n, \ \mathbf{E}_{-n} = \bar{\mathbf{E}}_n, \ \mathbf{H}_{-n} = \bar{\mathbf{H}}_n, \ \text{ and}$$

$$\mathbf{E}_{n+2} = \mathbf{F}_{n+1} = \mathbf{H}_n = \mathbf{0} \ \text{for} \ n > 2N_h$$

Note that $N_h = 0$ for isotropic rotors with more than two blades, whereas $N_h$ is selected for two-bladed rotors based on the required accuracy of the system matrix.

*Acknowledgements.* The support by the Danish Energy Agency through the EUDP-2011 II project 'Demonstration of Partial Pitch 2-Bladed Wind Turbine Performance' is gratefully acknowledged. The author would also like to thank Anders M. Hansen, Ilmar F. Santos, Torben J.

Larsen, Riccardo Riva, and project partners at Envision Energy for valuable discussions.





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

30  193.