# Peer review of "Modal dynamics of structures with bladed isotropic rotors and its complexity for 2-bladed rotors"

_Wind Energy Science, 2016_

## Author Comment (AC1) · 23 Aug 2016

Dear All,

I have noticed that I have made a typo in Equations (31) and (34). The index on the harmonic components of the periodic eigenvector in the summation should not be n-m but m-n.

Sorry for this mistake.

Best Regards, Morten
* * *

---

## Referee Comment (RC1) · D. Ossmann (Referee) · 10 Oct 2016

In my opinion the article presents a very interesting approach on periodic modelling the structural dynamics of wind turbines. The periodic dynamics for wind turbines are derived based on their Lagrangian. An eigen-mode analysis is presented using Hill's method for the periodic system. The method fully recovers the indeterminacy present for periodic models. The approach is applied to 2 and 3 bladed wind turbines.

The article is very well written technically and all the derivations are correct. It adds value to the field of modelling for wind turbines and uses a clear and correct borderline in the terms modes and degrees of freedom, which is often not correctly used in the today's literature. From my point of view there are some minor things the author might

take into account.

Specific comments:

*) First, I think the article is a bit verbosely written at some parts. Some sentences are too long and it would be easier for the reader if sentences are shorter. For example, the last sentences in the abstract has over 60 words. From a readability point of view this is too much. There are other examples in the article where readability could be improved (for example, first sentence in 4.4. is also rather long). I think going over it one more time focusing on that will help.

*) First sentence on page 2: Here the author could be more precise. For readers not fully familiar with the transformation of the modes onto the fixed frame it would be helpful pointing out that each SET of modes of the individual blades (e.g., flapwise, edgewise, etc) is transferred to symmetric, (anti-symmetric), regressive and progressive modes.

*) End of section 2.1. a bit more physical insight could be given on the individual meaning of the matrices to guide the reader through the derivations a bit better.

*) It is probably worth having on or two sentences on the Largrangian modelling approach, which will help starting section 2.1. from a readability point of view.

*) The Model Assurance Criterion at the end of Section 3.2. could be explained shortly or referenced.

Result section:

*) To be honest it took me quite some time to understand figure 5-15. Specifically I got confused with the numbering of the modes 2-12 and how they relate to diagram 4. I think it is not straightforward to the reader that if the modal amplitude in on mode is the highest, it gets named after that mode and ends up as name tag in Figure 4. Probably a table would help here.

*) How is the mentioned naming decided on, as some modes change their highest contribution in the periodic eigen-vector (as in Figure 8).

*) Can the scaling factor of 10 for the drive train be physically be motivated?

*) The author mentions a threshold of 10% when a component of the eigenvector is plotted/considered or not. However this might change with rotor speed. How is this handled?

*) In the result section I do not understand why actually N=3 has chosen for the three bladed rotor if the 3rd harmonics seems to be close to 0 anyway. Further, for the 2 bladed rotor the decrease is linear (in the log scale), so it is not immediately obvious why N=7 is chosen and not 9 or 5 or something else. I really like the figure and the discussion in 4.3. as it explains why the Coleman works that well for a three bladed rotor (dominance of the N=1 term).

*) Figure 16 and Line 11 on p 26: "The tower fore-aft mode in Figure 16 couples again with ±1/rev asymmetric rotor modes…." Shouldn't that be a coupling of DoFs, as a mode per definition is decoupled (orthogonal) from all the other modes?!

*) I really enjoyed reading section 4.6. Probably a bit more weight can be put on this outcome as I think it isa major contribution and handles some misunderstandings in common literature.

*) From my experience (and from the results in the paper) the Coleman transformation works pretty well when the rotors are isotropic, so it would be interesting to see - and add value to the approach - what happens if applied to higher level of anti-isomorphic characteristics for a 3-bladed rotor, for example on the blades (however this might be future work).

Technical corrections:

*) Typos:

Page 3, Line 10: I think it should be "can be used to decompose" instead of "decomposed" Page 9, Line 2: "with more than two" instead of "mode than two"

---

## Referee Comment (RC2) · V.A. Riziotis (Referee) · 11 Oct 2016

The paper compares the modal dynamics of two and three bladed turbines. The analysis is performed under a unified context based on the application of Hill's method. The paper is very well written and the key findings are highlighted in section 4.6 where the two types of rotors are compared (2 bladed vs. 3 bladed). I very much enjoyed reading it. One thing that could be missing is perhaps an example with a non isotropic 3 bladed rotor. I don't know if there is any space left since the paper is already quite long and seems to be out of the scope according to the title.

Some general comments:

[Figure]

Name convention for the modes is different in Figure 4 and the following figures. In figures 5-23 the author uses a number code. Although it is clearly explained in the text given that the figures are self explained (and this is nice) it would be better to use the same name convention (that describes the shape of the mode) also in plots 5-23.

Although the results of the analysis for the 3 bladed rotor are well expected, presentation of these results is found necessary in order highlight the differences between the two types of rotors. However, given the size of the paper (already 37 pages) a suggestion to the author would be to reduce slightly this part by presenting modal results for fewer modes and discuss the rest. Another idea could be to remove the 3D plotting of the modal displacements (at least in some of the plots) since it does not offer much. The picture is made clear already with the 2D plots I guess.

In section 4.4 the sentence starting "Looking at all figures" should be rephrased. After reading several times I understood the point the author wants to make but it is definitely not clear from first reading. I think the author tries to say something very obvious but in a rather complex way.

Some editorial changes,

Page 3, below line 15 "a 2 bladed turbine do not have" Page 5, below eq. 11 "DOFs are order as" Page 9, below eq 30, use upper case symbol on Nd Page 18, below line 10, "in close agreement of.."

---

## Author Comment (AC2) · 26 Oct 2016

Thanks for the many good comments and questions. I have answered them below and I have also uploaded a PDF with a latexdiff of the original submission and the suggested final submission.

*) First, I think the article is a bit verbosely written at some parts. Some sentences are too long and it would be easier for the reader if sentences are shorter. For example, the last sentences in the abstract has over 60 words. From a readability point of view this is too much. There are other examples in the article where readability could be improved (for example, first sentence in 4.4. is also rather long). I think going over it one more time focusing on that will help.

I agree and I have made changes to the abstract and gone through the manuscript with that in mind.

*) First sentence on page 2: Here the author could be more precise. For readers not fully familiar with the transformation of the modes onto the fixed frame it would be helpful pointing out that each SET of modes of the individual blades (e.g., flapwise, edgewise, etc) is transferred to symmetric, (anti-symmetric), regressive and progressive modes.

I agree and I have changed the sentence to: "The rotor modes of the 3-bladed isotropic rotor consist of a symmetric mode and two whirling modes for each blade mode (e.g. edgewise/flapwise bending and torsion). In the whirling modes, the order of blade vibration describes a backward (regressive) and a forward (progressive) whirling direction relative to the rotor rotation."

*) End of section 2.1 a bit more physical insight could be given on the individual meaning of the matrices to guide the reader through the derivations a bit better.

I recognize that a newcomer to structural modelling of rotary systems Section 2.1 may be a mouthful, but due to space issues I have shortened the section to an absolute minimum.

*) It is probably worth having on or two sentences on the Largrangian modelling approach, which will help starting section 2.1 from a readability point of view.

I originally had half a page more on the Lagrange equation and its linearization, but I shortened the text due to space issues. Instead I included a reference to a text book on the matter.

*) The Model Assurance Criterion at the end of Section 3.2 could be explained shortly or referenced.

I have added a reference to Allemang, R. J. (2003). The modal assurance criterion - Twenty years of use and abuse. Sound and Vibration, 37(8), 14–23.

[Figure]

*) To be honest it took me quite some time to understand figure 5-15. Specifically I got confused with the numbering of the modes 2-12 and how they relate to diagram 4. I think it is not straightforward to the reader that if the modal amplitude in on mode is the highest, it gets named after that mode and ends up as name tag in Figure 4. Probably a table would help here.

I totally agree and I have removed the reference to mode numbers, which did not make sense.

*) How is the mentioned naming decided on, as some modes change their highest contribution in the periodic eigen-vector (as in Figure 8).

Yes, the modes may have mode shape cross-overs as for the symmetric flap and FW flap in Figures 8 and 9. I have changed the introduction of this procedure to "The naming of the modes shown in Figure 4 is deduced from the harmonic components that dominate the periodic mode shapes observed in Figures 5 – 15 across a large part of the rotor speed range".

*) Can the scaling factor of 10 for the drive train be physically be motivated?

The scaling of tower top translations and drivetrain rotation angles is done to highly these components which can be considerably smaller than the blade deflections. I have changed the related text to "The generator rotation and shaft torsion angles are multiplied by 10 for better scaling of these small angles compared to blade deflections. Scaling of the tower translations are also applied to show weak couplings for some modes."

*) The author mentions a threshold of 10% when a component of the eigenvector is plotted/considered or not. However this might change with rotor speed. How is this handled?

I have changed the text introducing the threshold to: "Only amplitudes higher than 10 % of the overall maximum amplitude across all rotor speeds are plotted."

[Figure]

*) In the result section I do not understand why actually N=3 has chosen for the three bladed rotor if the 3rd harmonics seems to be close to 0 anyway. Further, for the 2 bladed rotor the decrease is linear (in the log scale), so it is not immediately obvious why N=7 is chosen and not 9 or 5 or something else. I really like the figure and the discussion in 4.3. as it explains why the Coleman works that well for a three bladed rotor (dominance of the N=1 term).

You are right. I have tried to mathematically prove that the third harmonic terms of the system matrix of an isotropic 3-bladed rotor are identical zero, but it requires that one looks deeply into the generic nature of the second order matrices (mass, gyro and stiffness). I will leave it open for later work or let others prove it.

I have truncated the Fourier series of the system matrix for the 2-bladed turbine at N=7 after looking at the modal solutions for other N values. I have not included this convergence study in the manuscript due to space issues. There are no significant improvement for N=9, but N=5 there is a small change in modal solutions.

*) Figure 16 and Line 11 on p 26: "The tower fore-aft mode in Figure 16 couples again with 1/rev asymmetric rotor modes: : :." Shouldn't that be a coupling of DoFs, as a mode per definition is decoupled (orthogonal) from all the other modes?!

You are right. I have changed the sentence to "The tower fore-aft mode in Figure 16 couples again with $\pm$1/rev asymmetric rotor motion as for the 3-bladed turbine" and looked for similar issues in the manuscript.

*) I really enjoyed reading section 4.6. Probably a bit more weight can be put on this outcome as I think it is a major contribution and handles some misunderstandings in common literature.

I agree. I am planning to submit a paper only on the difference of two and three bladed turbines in the near future. This paper is meant as an introduction of the theories.

*) From my experience (and from the results in the paper) the Coleman transformation

works pretty well when the rotors are isotropic, so it would be interesting to see – and add value to the approach - what happens if applied to higher level of anti-isomorphic characteristics for a 3-bladed rotor, for example on the blades (however this might be future work).

I agree, but I will leave that important topic for the next paper.

*) Typos: Page 3, Line 10: I think it should be "can be used to decompose" instead of "decomposed" Page 9, Line 2: "with more than two" instead of "mode than two"

Thanks. I have made the corrections.

Please also note the supplement to this comment:
http://www.wind-energ-sci-discuss.net/wes-2016-27/wes-2016-27-AC2-supplement.pdf

---

## Author Comment (AC3) · 26 Oct 2016

Thanks for the many good comments and questions. I have answered them below and I have also uploaded a PDF with a latexdiff of the original submission and the suggested final submission.

\*) One thing that could be missing is perhaps an example with a non-isotropic 3-bladed rotor.

I agree, but I will leave that important topic for the next paper. As you also write the paper is already very long. I have focused on the description of the methods herein.

\*) Name convention for the modes is different in Figure 4 and the following figures. In

figures 5-23 the author uses a number code. Although it is clearly explained in the text given that the figures are self explained (and this is nice) it would be better to use the same name convention (that describes the shape of the mode) also in plots 5-23.

I have removed the reference to mode numbers, which did not make sense. The modes are now only referred to by their name, which is this procedure: "The naming of the modes shown in Figure 4 is deduced from the harmonic components that dominate the periodic mode shapes observed in Figures 5 – 15 across a large part of the rotor speed range".

\*) Although the results of the analysis for the 3 bladed rotor are well expected, presentation of these results is found necessary in order highlight the differences between the two types of rotors. However, given the size of the paper (already 37 pages) a suggestion to the author would be to reduce slightly this part by presenting modal results for fewer modes and discuss the rest. Another idea could be to remove the 3D plotting of the modal displacements (at least in some of the plots) since it does not offer much. The picture is made clear already with the 2D plots I guess.

I have gone through the text again and shorten it in places. You are one of the experts on the area of wind turbine dynamics; I am afraid that if I would write a short text appealing to your expert knowledge then I may lose new readers. I would like to avoid removing the 3D plots because they tell us which harmonic components in the other plots have the largest amplitudes.

\*) In section 4.4 the sentence starting "Looking at all figures" should be rephrased. After reading several times I understood the point the author wants to make but it is definitely not clear from first reading. I think the author tries to say something very obvious but in a rather complex way.

I agree and I have also rephrased this section.

\*) Some editorial changes, Page 3, below line 15 "a 2 bladed turbine do not have" Page

5, below eq. 11 "DOFs are order as" Page 9, below eq 30, use upper case symbol on Nd Page 18, below line 10, "in close agreement of.."

Thanks. I have made the corrections.

Please also note the supplement to this comment:
http://www.wind-energ-sci-discuss.net/wes-2016-27/wes-2016-27-AC3-supplement.pdf

**Supplement:**

[revised manuscript text omitted]